# H2IL-MBOM: A HIERARCHICAL WORLD MODEL INTEGRATING INTENT AND LATENT STRATEGY AS OPPONENT MODELING IN MULTI-UAV GAME

## ABSTRACT

In the mixed cooperative-competitive scenario, the uncertain decisions of agents on both sides not only render learning non-stationary but also pose a threat to each other's security. Existing methods either predict policy beliefs based on opponents' interactive actions, goals, and rewards or predict trajectories and intents solely from local historical observations. However, the above private information is unavailable and these methods neglect the underlying dynamics of the environment and relationship between intentions, latent strategies, actions, and trajectories for both sides. To address these challenges, we propose a Hierarchical Interactive Intent-Latent-Strategy-Aware World Model based Opponent Model (H2IL-MBOM) and the Mutual Self-Observed Adversary Reasoning PPO (MSOAR-PPO) to enables both parties to dynamically and interactively predict multiple intentions and latent strategies, along with their trajectories based on self observation. Concretely, the high-level world model fuses related observations regarding opponents and multi-learnable intention queries to anticipate future intentions and trajectories of opponents and incorporate anticipated intentions into the low-level world model to infer how opponents' latent strategies react and their influence on the trajectories of cooperative agents. We validate the effectiveness of the method and demonstrate its superior performance through comparisons with state-of-the-art model-free reinforcement learning and opponent modeling methods in more challenging settings involving multi-agent close-range air-combat environments with missiles.

## 1 INTRODUCTION

In multi-agent environments, numerous agents interact and learn concurrently, influencing each other's transition dynamics and rendering the environments non-stationary. Additionally, adapting to the ever-changing and unknown policies of other agents introduces non-stationary strategies, posing challenges for policy improvement. Importantly, navigating interactions with these unknown opponent policies can jeopardize allies' safety and limit performance, particularly in competitive or mixed cooperative-competitive scenarios. Therefore, modeling opponents behavior or reasoning their intentions is crucial for ensuring safety and achieving supremacy during decision-making processes.

Opponent modeling and intent reasoning are integral components of Theory of Mind (ToM), granting agents the ability to infer opponents' mental states, including preferences, styles, desires, beliefs, goals, intentions, latent strategies, or behaviors. Some approaches reconstructed policy beliefs and predicted opponents' actions based on the assumption that opponents' behaviors are known from experience. Other works focused on extrapolating adversary strategies/intentions and trajectories from local observation experience without prior assumptions. However, the assumptions underlying the former method are unrealistic in the real world, as opponents typically do not disclose their actions, strategies, or intentions. Furthermore, methods that directly estimate opponent latent strategies based on their trajectories and predict future trajectories encounter challenges with a large state space and overlook the environment's underlying dynamics, failing to comprehend how intentions, latent strategies, and actions dynamically interact for both sides. Specifically, they remain unclear how estimated opponent intentions will influence their latent strategy, how ego agents will react to opponents' intentions and latent strategies, and consequently, how future intentions, latent strategies, and trajectories of both sides will be affected. In other words, they lack insight into how opponents'

latent strategies, influenced by time-varying intentions, will impact allies' trajectories and struggle to continuously reason about the opponents' evolving intentions and strategies due to the influence of future trajectories during the next iteration. Moreover, most opponent modeling methods only train and test in matrix game, differential game, cooperative navigation, triangle game, and StarCraftII, and so on, they have not yet trained in such high dynamic air-combat environments.

Developing a human-like opponent model or intent reasoning model inevitably presents challenges. Maintaining a multi-hypothesis intention and strategy for opponents with advanced cognitive abilities in dynamic and complex competitive-cooperative scenarios, adapting to a variable number of adversaries with changing intentions, and dealing with the resulting uncertainty in strategy estimation are necessary.

Main contributions: To address these challenges and fill the gap in opponent modeling without accessing any private information of opponents in the field of air-combat, drawing inspiration from the brain's cognitive process, we propose a Hierarchical Interactive Intent-Latent-Strategy-Aware World Model based Opponent Model (H2IL-MBOM), specially a Hypernetwork-based Hierarchical Dynamic Dependence Transformer State Space Model (HyperHD2TSSM), and a Mutual Self-Observed Adversary Reasoning PPO (MSOAR-PPO) for reasoning about opponents' multi-intentions and latent strategies without accessing any private information of opponents. Our contributions are five-fold:

1. We derive a hierarchical variance inference and construct a novel hierarchical world model based opponent model for dynamically and interactively learning and inferring multi-intentions, latent strategies, and trajectories of opponents and allies without accessing any private information of opponents.

2. The developed HyperHD2TSSM allows for the establishment of an interactive transition model for all agents without increasing parameters, enhancing adaptability and scalability. Moreover, the latent weights generated by the hypernetwork of each agent at each time step compress historical information about opponents' mental states reasoned by adjacent agents, achieving $O(1)$ complexity as in recurrent state space models (RSSM) (Hafner et al. (2019b)) while maintaining parallel training as in transformer state space model (TSSM) (Chen et al. (2022)).

3. A novel approach has been devised to more accurately predict the impact of opponents' intentions on their strategies without clear intentions and latent strategies candidates, effectively extracting the distribution of opponents' behavioral patterns, and thereby enhancing interpretability.

4. The developed MSORA-PPO equipped with the H2IL-MBOM can infer adversarial strategies and intentions based on self-observation in real time, facilitating rapid adaptation to changes in multiple opponents' intentions and strategies and addressing the non-stationarity issue caused by opponents' continuous learning.

5. To the best of my knowledge, this is the first work to integrate intent and latent strategy into the world model as opponent modeling, validated in a multi-agent close air-combat game with missiles in a gym-jsbsim environment, promoting theoretical development and application in the field of air-combat. The results demonstrate superior performance as our method can capture changing behavior patterns of opponents and possesses generalization ability.

## 2 RELATED WORK

**Opponent modeling.** Opponent modeling involves inferring opponents' mental states, such as desires, goals, actions, beliefs, and intentions, to address non-stationarity and gain supremacy when facing unknown and changeable opponent policies. Previous methods like DPN-BPR+ (Zheng et al. (2018)) and ToMoP (Yang et al. (2018)) offer strategies to detect and reuse opponent strategies, albeit struggling with multiple continuously evolving opponents. Methods like RFM (Tacchetti et al. (2018)), P-BIT (Tian et al. (2020)), ROMMEO (Tian et al. (2019)), TOM (Rabinowitz et al. (2018)), SOM (Raileanu et al. (2018)), LeMOL (Davies et al. (2020)), and TDOM (Tian et al. (2023)) utilize opponents' observations, actions, or rewards to infer their goals or beliefs, though lacking access to opponents' private information. PR2 (Wen et al. (2019)) and GR2 (Wen et al. (2021)) propose multi-agent probabilistic recursive inference but cannot simultaneously learn policies for agents. GrAMMI (Ye et al. (2023)) emphasizes multi-hypothesis belief over opponents and uses

mutual information theory to predict opponent behaviors. Although Busch (Busch et al. (2022)), Wu et.al. (Wu et al. (2023)) and Xie et.al. (Xie et al. (2021)) predicted adversary' incentive, strategies or trajectories, they lack insight into how time-varying intentions affect opponent strategies and trajectories of all agents. (Yu et al. (2022); Zhang et al. (2021)) predicted opponent' actions based on world model, whereas they still required real opponents' actions as learning labels. Further more, they have not yet trained in such high dynamic air-combat environments. Our approach differs by inferring intentions of multiple opponents from their historical and current observations without accessing private information in the field of air-combat. We infer opponent latent strategies from cooperative neighbors' observations and use intent to understand their impact on strategies and collaborative agent trajectories, dynamically inferring multi-evolving opponent intentions and latent strategies.

**World Model.** Current single-agent world models include like MBPO (Kaiser et al. (2019)), DreamerV1-V3 (Hafner et al. (2019a; 2020; 2023)) based on RSSM (Hafner et al. (2019b)), TSSM developed by (Chen et al. (2022)), and graphical state space model (GSSM) developed by (Wang & Van Hoof (2022)). Some extend single-agent models to multi-agent models, categorized as centralized (Willemsen et al. (2021)) or decentralized (Xu et al. (2022); Hu et al. (2021); Egorov & Shpilman (2022)). However, these models become cumbersome as the number of agents increases and make independent predictions, and mainly focused on cooperative navigation in environments such as MPE or mazes. On the contrary, we focus on how to build an interactive multi-agent world model that can make interactive prediction and own different ways of information compression in the multi-uav adversary environment. Specially, our model allocates diverse latent weights to each agent, dynamically adjusting them based on neighboring agents' preceding latent states, enabling spatiotemporal sequence forecasting. This characteristic allows it to establish transition models and interactive predictions for each agent without increasing parameters, enhancing adaptability and scalability compared to centralized and decentralized models.

## 3 METHODOLOGY

**Problem Statement.** We consider mixed cooperative-competitive scenarios involving $N >= 2$ agents. Each agent operates based on its intentions and strategies while interacting with others without accessing private information of competitive agents, such as learning algorithms, actions, rewards, goals, and incentives. These private details of opponents, including adversaries and missiles, remain diverse, changeable, and unknown to cooperative agents. In this study, we aim to understand opponents' mental states by constructing H2IL-MBOM models from their perspectives, and using these these predictions along with observations to inform decision-makings. Therefore, we have two objectives. The Markov decision process comprises a tuple $\langle N, n, M, m, S, A, O, Z, H, R, \gamma \rangle$ where $N$ and $n$ are numbers of cooperative agents and observable cooperative neighbors, respectively; $M$ and $m$ are numbers of opponents and observable opponents, respectively; $S$ is the state sets, $A = \{A_i\}_{i=1}^N, O = \{O_{opp}, O_c\} = \{O_i\}_{i=1}^N = \{O_{opp,i}, O_{c,i}\}_{i=1}^N$ are the action sets and observation sets relative to opponents $O_{opp}$ and cooperative neighbors $O_c$. $z = \{z_I, z_L\} = \{z_i\}_{i=1}^N = \{z_{I,i}, z_{L,i}\}_{i=1}^N$ are incentive representations, which consist of intentions $z_I$ and latent strategies $z_L$. $H = \{H_{opp,t}, H_{c,t}\} = \{\{O_{opp,i,t}\}_{t=t_0,...,t-1}^{i=1,...,N}, \{O_{c,i,t}\}_{t=t_0,...,t-1}^{i=1,...,N}\}$ signifies the historical observations of agents; and $R, \gamma$ are rewards and discount factor, respectively. The first objective is to maximize the expected return $E_\pi \left[ \sum_{t=0}^{\infty} \gamma^t R_t(s_t, \{a_{i,t} \sim \pi(|o_{i,t}, z_{I,i,t}, z_{L,i,t})\}_{i=1}^N, s_{t+1}) \right]$, and the second objective involves updating reasoned intentions and latent strategies based on future ground-truth incentive representations

**Core idea.** When facing multiple opponents, humans typically begin by inferring multi-intentions through analyzing historical and current observations, consider how estimated intentions of opponents affect their strategies, and contemplate their next moves anticipating opponent intentions and strategies. This allows them to envision future interaction states, predict emerging intentions and strategies, and perpetuate a cycle of strategic anticipation and adaptation. Inspired by these cognitive processes, we constructed high-level dynamic intent-aware representation fusion (HDIRF) model that consists of a high-level history transformer encoder (H2TE)-multi-intention transformer decoder (MITD) to parse initial multi-intention queries, a low-level dynamic latent-strategy-aware representation fusion (LDLRF) model that consists of a low-level history transformer encoder (LHTE)-multi-latent-strategy transformer decoder (MLTD) to predict multi-latent strategies influenced by estimated intentions,

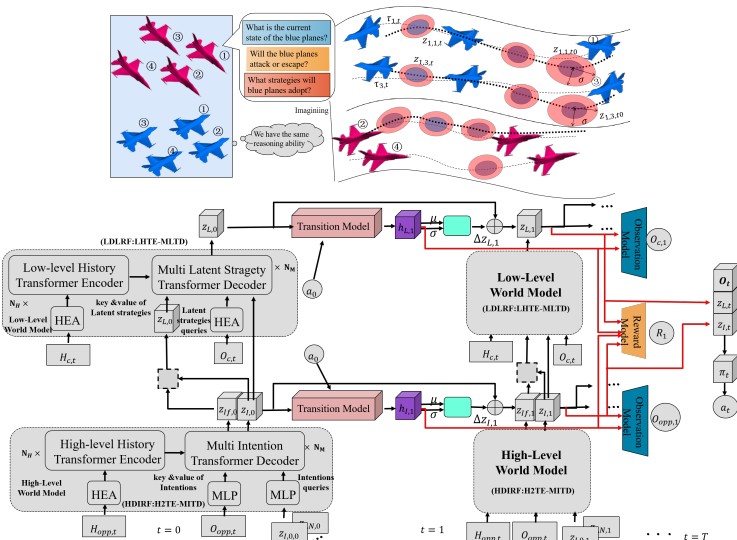

Figure 1: Overview of the H2IL-MBOM, which comprises high-level world model and low-level model.

and an interactive hypernetwork-based joint latent gated transformer (HJLGT) as transition model to interactively infer future mental states of opponents and reconstruct trajectories of opponents and cooperative agents, endowing agents with the ability to simulate the brain. The hierarchical variance inference is detailed in Section 3.2, and the dynamic fusion mechanisms is detailed in Section 3.3.

## 3.1 MUTUAL REACTION AND INFLUENCE BETWEEN MENTAL STATES, ACTIONS, AND TRAJECTORIES

During each interaction, the red and blue teams serve as each other's opponents, and both teams of agents progress through a series of stages. These stages encompass the influence of the opponents' intentions on their strategies during the interaction; the influence of these intentions on the opponents' trajectories, as well as the impact of the opponents' strategies on the trajectories of the alliance throughout the interaction; and the subsequent influence of these interactive trajectories on both intentions and strategies in the repeated interactions.

**Stage one: the influence of the opponents' intentions on their strategies during the interaction.** Following the $t-1$ iteration, both teams of agents are capable of gathering their respective histories of interaction observations. At this juncture, an intelligent agent ought to be able to discern the intentions of its opponent, as well as the strategies that the opponent may employ in response to varying intentions.

**Stage two: the influence of these intentions on the opponents' trajectories and the opponents' strategies on the trajectories of the alliance throughout the interaction.** From the perspective of each intelligent agent, actions are taken in response to the predicted opponents' intentions and strategies, and the opponent's mental states evolve in reaction to the behavior of the alliance agents. This mutual response is contingent upon interactions with different agents, leading to uncertain transition probabilities $\mathrm{P}(O_{i,t}|O_{i,t-1}, a_{i,t-1}, z_{i,t-1})$. Moreover, the opponent's evolving intention directly reflects changes in the opponents' trajectories, while their evolving strategy influences the trajectories of the alliance agents.

**Stage three: the subsequent influence of the interactive trajectories on both intentions and strategies in the repeated interactions.** After the $t-th$ interaction, the cooperative agents update their policies for the next moment based on the affected trajectory observations, historical observations, and the opponents' intentions and strategies inferred from these observations. At the same time, the opponents also update their intentions and strategies to respond to the policies of the cooperative agents. After the interaction of $T$ timesteps, both agents experience trajectories $\{(O_{i,t}, a_{i,t}, r_{i,t}), ..., (O_{i,t+T}, a_{i,t+T}, r_{i,t+T})\}$ with the goal of achieving long-term rewards.

## 3.2 Hierarchical Variance Inference

The three stages illustrate that the opponent's intentions and strategies evolve alongside the transfer dynamics. Insights from stages one and two reveal that different opponent intentions result in distinct strategies, and that intentions and strategies, in turn, reflect changes in the opponents' trajectories and their impact on the trajectories of cooperative agents. Consequently, both agents can partition their local current and historical observations into those related to the opponents and those related to cooperative agents. They can then estimate the opponents' intentions using their current and historical observations of the opponents. Additionally, the agents estimate the opponents' strategies based on their current and historical observations of the cooperative agents, combined with their estimates of the opponents' intentions.

However, the actions, intentions, and strategies of the opponents cannot be directly observed. How can we predict these representations without explicit labels about the opponents' private information? One straightforward approach is to use an autoencoder to directly learn representations of historical and current observations and predict future trajectory sequences, akin to the work in Qi & Zhu (2018). Drawing from insights in stages two and three, the actions, intentions, and strategies of both parties interact, jointly determining the future trajectory. This trajectory, in turn, influences the evolving intentions, strategies, and actions of both parties. In other words, the future trajectory is influenced not only by the opponent's current mental state but also by their evolving intentions and strategies. Therefore, capturing trajectory changes in dynamic adversarial environments using an autoencoder-based approach is inherently challenging.

To implicitly capture the mutual reactions and impacts between the opponent's intentions, strategies, behaviors, and the trajectories of both sides, based on insights from the three stages, we recognize the necessity of introducing transition models alongside actions. The teams of agents interactively predict and capture the mutual reactions between the opponents' intentions, strategies, and behaviors using the transition model at each time step. As highlighted in stage two, the mutual reactions between the opponents' intentions and the actions of the team are reflected in trajectory transitions related to observations for the opponents. Similarly, the mutual reactions between the opponents' latent strategies and the team's actions result in transitions of trajectory sequences related to observations for the cooperative agents.

Therefore, we approximate higher-level and lower-level transition models using $p_{\psi_I}$ and $p_{\psi_L}$, respectively, and use them to continuously predict the changes in the opponent's intentions and strategies, as well as the hidden states $(h_{I,i,t}, h_{L,i,t})$ that compress historical information about predictions of the opponent's intentions and strategies reasoned by themselves and other cooperative agents. Notably, these hidden states differ from those in recurrent neural networks; they are deterministically generated by the hypernetwork for each agent at each time step, specifically the output of the deterministic model within the transition model, as detailed in Figure 5. As a result, each prediction at any given time has a complexity of $O(1)$, avoiding the $O(T)$ complexity associated with explicitly requiring history up to time $T$ as in TSSM. Additionally, in this process, we integrate the predicted intentions and strategies of the opponent, along with the evolutionary history (hidden states) of these mental representations, to reconstruct the affected trajectories of the opponent and cooperative agents. Thus, the reconstruction probability is related to the prior estimates of the hidden states, i.e., $p_{\theta_I}(O_{opp,i,t}|z_{I,i,t}, h_{I,i,t})$ and $p_{\theta_L}(O_{c,i,t}|z_{L,i,t}, h_{L,i,t})$, and the future trajectories further influence the actions and mental representations of both sides in the next time step. Overall, during the interaction, the trajectories continually influence or induce changes and updates in the opponents' mental representations. Gradually, the agents on both sides maximize long-term rewards by learning the evolving mental states of the respective opponents over multiple steps of interaction. The reasoning and learning of the entire world model are as follows:

As shown in Figure 1, allies and opponents are equipped with the same H2IL-MBOM, a Hyper-HD2TSSM for estimating mental states of each other. For example, collaborative agents utilize this model for opponent modeling with historical observations sequence $H_{opp,t}$ and the current observation $O_{opp,i,t}$ regarding opponents at each time, and vice versa. $O_{opp,i,t} = \{O_{i,j,t}\}_{j=1,...,m}$ represents the observations regarding $m$ opponents within the observation scope of the agent $i$, and $H_{opp,t} = \{O_{opp,i,t}\}_{t=t_0,...,t-1}^{i=1,...,N}$, where the agent $i$ employs the H2TE-MITD to approximate the High-level posterior $q_{\phi_I}(z_{I,i,t}|H_{opp,t}, O_{opp,i,t})$ to estimate multi-intent queries $z_{I,i,t}$ of opponents. It also uses a deterministic model $HJLGT_I$ and a Gaussian stochastic model as approximations of the

high-level prior $p_{\psi_I}(z_{I,i,t}|z_{I,i,t-1}, z_{I,n_i,t-1}, a_{i,t-1}, a_{n_i,t-1})$ related to the speculation of the intent $z_{I,n_i,t-1}$ of the $n$ neighbors $n_i = \{1,..,n\}_{\neq i}$ of the agent $i$ towards adversaries and the actions of the neighbors of the agent $i$ to directly infer future multi-intent queries. Furthermore, it estimates the observation model $p_{\theta_I}(O_{opp,i,t}|z_{I,i,t}, h_{I,i,t})$ with hidden states $h_{I,i,t}$ to predict trajectory states of opponents by considering the influence of current and historical intentions, which reveals that intentions reflect the variations of opponents' trajectories. At the low-level world model, the LHTE-MLTD approximates the low-level posterior $q_{\phi_L}(z_{L,i,t}|H_{c,t}, O_{c,i,t}, z_{I,i,t})$ to estimate multi-latent strategy queries $z_{L,i,t}$ from the historical observations sequence $H_{c,i,t} = \{O_{c,i,t}\}_{t=t_0,...,t-1}^{i=1,...,N}$ to allies, current observation $O_{c,i,t} = \{O_{i,l,t}\}_{l=1,...,n(l\neq i)}$ to cooperative neighbors, and current intent queries $z_{I,i,t}$, which considers the impact of intentions on latent strategies. Similarly, it utilizes a deterministic model $HJLGT_L$ and a Gaussian stochastic model to approximate low-level prior $p_{\psi_L}(z_{L,i,t}|z_{L,i,t-1}, z_{L,n_i,t-1}, a_{i,t-1}, a_{n_i,t-1}, z_{I,i,t})$ based on the reasoning of the latent strategies $z_{L,n_i,t-1}$ of the $n$ neighbors of the agent $i$ towards the adversaries, the actions of the neighbors of the agent $i$, and predicted intent queries $z_{I,i,t}$. The observation model $p_{\theta_L}(O_{c,i,t}|z_{L,i,t}, h_{L,i,t})$ with hidden states $h_{L,i,t}$ predicts trajectories of cooperative neighbors driven by the current and historical estimated latent strategies $z_{L,i,t}$ of opponents, which reveals the influence of opponents' latent strategies on cooperative neighbors' trajectories. After estimating $z_{I,i,t}, z_{L,i,t}$ at each step, the agent $i$ can make the decision $a_{i,t} = \pi(O_{opp,i,t}, O_{c,i,t}, z_{I,i,t}, z_{L,i,t})$ and infer rewards $p_{\theta_r}(r_{i,t}|z_{I,i,t}, h_{I,i,t}, z_{L,i,t}, h_{L,i,t})$. The hierarchical evidence lower bound (HELBO) is derived by Jensen's inequality as follows:

$$
\begin{aligned}
&\log p(O_{opp,1:N,1:T}, O_{c,1:N,1:T}, a_{1:N,1:T}, h_{I,1:N,1:T}, z_{I,1:N,1:T}, h_{L,1:N,1:T}, z_{L,1:N,1:T}) \\
&= \log E_{q(z_{1:N,1:T}|H_{1:T}, O_{1:N,1:T})} \left[ \frac{p(O_{opp,1:N,1:T}, O_{c,1:N,1:T}, a_{1:N,1:T}, h_{I,1:N,1:T}, z_{I,1:N,1:T}, h_{L,1:N,1:T}, z_{L,1:N,1:T})}{q(z_{1:N,1:T}|H_{1:T}, O_{1:N,1:T})} \right] \\
&\geq E_{q(z_{1:N,1:T}|H_{1:T}, O_{1:N,1:T})} \log \left[ \frac{p(O_{opp,1:N,1:T}, O_{c,1:N,1:T}, a_{1:N,1:T}, h_{I,1:N,1:T}, z_{I,1:N,1:T}, h_{L,1:N,1:T}, z_{L,1:N,1:T})}{q(z_{1:N,1:T}|H_{1:T}, O_{1:N,1:T})} \right] \\
&= \sum_{t=1}^{T} \sum_{i=1}^{N} \begin{array}{l} E_{q(z_{I,i,1:t}|H_{opp,1:t}, O_{opp,i,1:t})}(\log[p(O_{opp,i,t}|h_{I,i,t}, z_{I,i,t})]) + E_{q(z_{L,i,1:t}|H_{c,i,1:t}, O_{c,i,1:t}, z_{I,i,1:t})} \\ (\log[p(O_{c,i,t}|h_{L,i,t}, z_{L,i,t})]) + E_{q(z_{I,i,1:t}|H_{opp,1:t}, O_{opp,i,1:t})q(z_{L,i,1:t}|H_{c,1:t}, O_{c,i,1:t}, z_{I,i,1:t})} \\ \log[p(a_{i,t}|O_{opp,i,t}, O_{c,i,t}, z_{I,i,t}, z_{L,i,t})] - E_{q(z_{I,i,1:t}|H_{opp,1:t}, O_{opp,i,1:t})} KL(q(z_{I,i,t}|H_{opp,t}, O_{opp,i,t})|| \\ p(z_{I,i,t}|z_{I,i,t-1}, z_{I,n_i,t-1}, a_{i,t-1}, a_{n_i,t-1})) - E_{q(z_{L,i,1:t}|H_{c,1:t}, O_{c,i,1:t}, z_{I,i,1:t})} \\ KL(q(z_{L,i,t}|H_{c,t}, O_{c,i,t}, z_{I,i,t})||p(z_{L,i,t}|z_{L,i,t-1}, z_{L,n_i,t-1}, a_{i,t}, a_{n_i,t-1}, z_{I,i,t})) \end{array}
\end{aligned}
\tag{1}
$$

Please refer to Appendix A.8 for the derivation. The third term can be ignored due to the joint policy. In the process of learning to estimate the opponent's intentions and strategies, to reduce the cumulative error in predicting these intentions and strategies, we use two Kullback-Leibler (KL) divergence as loss functions to minimize both prior and posterior estimate of the hierarchical world model. Then, we reconstruct the observations regarding the opponents' trajectories and the trajectories of the cooperative agents using the posterior estimates and the prior hidden states, i.e., $E_{q(z_{I,i,1:t}|H_{opp,1:t}, O_{opp,i,1:t})}(\log[p(O_{opp,i,t}|h_{I,i,t}, z_{I,i,t})]) + E_{q(z_{L,i,1:t}|H_{c,i,1:t}, O_{c,i,1:t}, z_{I,i,1:t})}(\log[p(O_{c,i,t}|h_{L,i,t}, z_{L,i,t})])$. The prior estimates $h_{I,i,t}$ and $h_{I,i,t}$ plays a dual role; it not only impacts the predicted trajectory within the observation model but also affects the learning of the posterior estimate because of the reparameterization trick in the reconstruction loss. Consequently, the reconstructed trajectories and the learning of the posterior estimate are related to opponents' evolving mental states, which diverges from the learning manner of autoencoders. Given that the opponents' intentions at each step not only reflect changes in the opponents' trajectories but also influence the lower-level strategies and the trajectories of the cooperative agents, the intentions are updated through the hierarchical world model via two rounds of backpropagation. The comparison with RSSM, TSSM and our HyperHD2TSSM can be found in Appendix A.3. In addition, the transition model HJLGT can be found in Appendix A.4.

### 3.3 DYNAMIC FUSION MECHANISMS OF INTENTIONS AND LATENT STRATEGIES IN OPPONENT MODELING

Given the learning and reasoning capabilities of both sides, the historical trajectories exhibit variation across different episodes. Moreover, the mental states of the opponents also evolve over time with the dynamics of the game. Therefore, the H2IL-MBOM is designed to enable UAVs to dynamically predict opponents' evolving intentions and latent strategies over the most recent trajectories in a mixed cooperative-competitive scenario. The HDIRF is responsible for predicting the intentions of opponents based on historical and current observations. This is achieved through the integration of

a H2TE and a MITD. The H2TE captures the historical context relevant to the opponents, while the MITD decodes the intentions from the encoded historical context. This process involves the updating of intention feature queries via a self-attention mechanism, followed by a cross-attention mechanism that considers the assembled historical features of all cooperative agents' observations for the opponents. Additionally, A fusion module integrates the outputs of the mechanisms and the inferred intentions embeddings generated by the hypernetwork to produce a unified representation of the opponents' intentions, taking into account the interactions of intentions across agents. The initial intentions embeddings are generated from the initialized intentions through the hypernetwork, while subsequent intentions embeddings are derived from the intentions inferred in the previous layer. For scenarios where different agents face the same adversaries, the module can capture the interaction and assign different threat weights to each opponent's intention for each agent.

The LDLRF focuses on inferring latent strategies and understanding how these strategies respond to the predictions of opponents' intentions. It utilizes a LHTE and a MLTD. The LHTE encodes the historical observations of cooperative agents, and the MLTD decodes the latent strategies by considering the influence of opponents' intentions. The process includes a hypernetwork-based intention self-cross-attention module, which captures the impact of opponents' intentions, followed by a latent-strategy cross-attention module that aggregates historical features regarding all cooperative agents. Finally, a latent-strategy fusion module captures the interactions of latent strategies across agents. Similarly, the initial latent strategies embeddings are generated from the initialized latent strategies via the hypernetwork, while subsequent latent strategies embeddings are derived from the latent strategies inferred in the previous layer.

Moreover, the mental states are automatically generated by the respective hypernetworks, thereby handling changes in the number of opponents and the number of opponents' intentions and latent strategies.

By incorporating these hierarchical mechanisms, H2IL-MBOM facilitates a comprehensive understanding of the opponents' behavior, enabling drones to make informed decisions based on a global prediction of opponents' intentions and a precise capture of evolving latent strategies. The detailed calculations of HDIRF that consists of H2TE-MITD, and LDLRF that is composed of LHTE-MLTD can be also found in the Appendix A.5, and A.6.

### 3.4 IMPLEMENTATION

Two teams engage in independent policy learning, value learning, and world model learning based on their local observations due to the limitation of imperfect game, which differs from MAPPO. The world model, when given a tuple $(O_t, a_t, O_{t+1}, H_t, r)$, outputs mental states $z_I, z_L$ through a hierarchical model and reconstructs the trajectory states and rewards of teammates and opponents in the next time step. In the reinforcement learning module, the policy network and value network integrate their respective local observations and predicted mental states into the hypernetwork-based embedded attention (HEA) model, and are updated based on PPO. Here, we also use the whole world model that includes a reward model to generate three-step imaginations of trajectories, and then combine imagining trajectories and real trajectories to compute $J_\pi, J_V$. Specifically, we calculate the returns $G_1$ and $G_2$ for the actual trajectory and imaging trajectory respectively, and use the identical value function to calculate the values $V_1$ and $V_2$ for these trajectories. Finally, we calculate the advantages $A_1$ and $A_2$ of the trajectories by respective rewards and values. In policy learning, we update the policy using the advantages of these trajectories, and update the common value function using the returns $G_1$ and $G_2$ of the trajectories. It is worth noting that we do not learn another value function like Dreamer, which can ensure performance while reducing complexity. The pseudo code can be found in 1.

## 4 EXPERIMENTS

We validate our method in a mixed cooperative-competitive environment Gym-Jsbsim (Liu et al. (2022)), where we select F16 aircraft as the agents for the red and blue teams to engage in a 4 vs.4 dogfight scenario. We randomly initialize the initial state of the F16 in each episode, then compare our method with various methods, examine the learning performance and the efficacy of our method, and validate the necessity and contribution of designing modules. Further details regarding environment

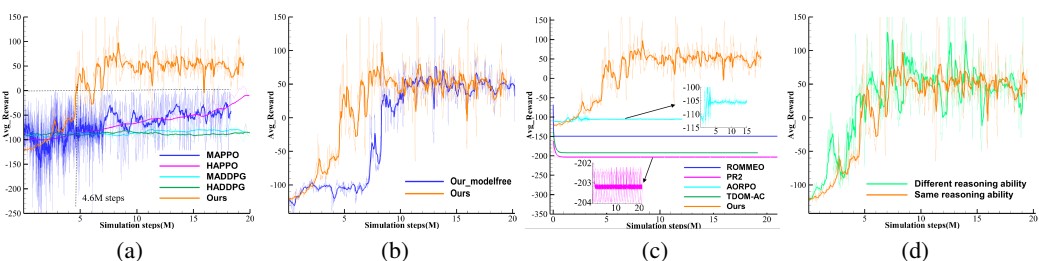

Figure 2: Performance comparison of various methods.

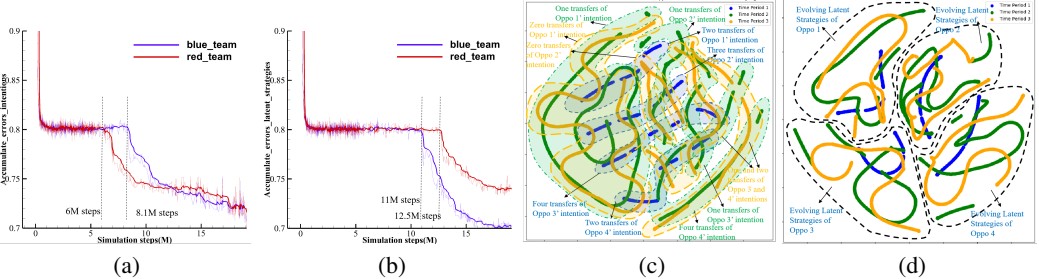

Figure 3: Accumulate error and t-SNE distribution of opponents' multiple intentions and latent strategies reasoned by Agent0 across three time periods. The total number of intention transitions observed for all opponents across various stages is 11, 7, and 3, respectively. In contrast, the low-level strategies employed by the opponents exhibit a more consistent and distinguishable performance.

setting, hyperparameters, and additional experiments are provided in Appendix A.2,A.11,and A.9–A.10, respectively. It is noting that we use two training methods as shown in Figure 2d instead of employing built-in AI for opponents: one is where both sides train own H2IL-MBOM and policy based on their respective local observations, respectively, and the other is where the blue side adopts the historical strategies of the red side, which is a self-play technique. They competed against opponents that were never encountered in the training phase.

## 4.1 COMPARISON WITH VARIOUS METHODS

For each algorithm, we use the same network architecture and hyperparameters as described in corresponding literature. To ensure fair comparison, we train these baseline algorithms under the same conditions such as initial conditions, same number of training steps.

**Comparison with model-free MARL.** We compare our method with CTDE MARL involving MAPPO (Yu et al. (2021)) and MADDPG (Lowe et al. (2017)), decentralized methods involving HAPPO and HADDPG (Zhong et al. (2024)), and our model-free version RL, in which MAPPO and HAPPO make random actions while MADDPG and HADDPG make deterministic actions. For each algorithm, we use the same network architecture and hyperparameters as described in corresponding literatures. To ensure fair comparison, we train these baseline algorithms under the same conditions such as initial conditions, same number of training steps. As illustrated in Figures 2a, other MARLs present negative rewards. MADDPG and HADDPG exhibit poor performance due to the dynamic and evolving behaviors of both parties, as deterministic actions are unable to adapt to such complexity. Consequently, both teams are susceptible to missile attacks from each other. While MAPPO and HAPPO have improved the situation, their performance fluctuates due to environmental non-stationarity, and rewards are less than zero. And this method has improved performance as illustrated in Figure 2b. Due to our method's ability to infer the opponent's intentions and strategies, we can overcome environmental non-stationarity issues even based on local observations, ensuring safety to the greatest extent possible, resulting in a positive reward of 50 with relatively minor fluctuations.

**Comparison with other opponent modeling methods.** We compared the recent opponent modeling approaches: ROMMEO, PR2, TDOM-AC and AORPO. As presented in Figure 2c, these methods, these approaches exhibit fluctuations in negative rewards, with the reward of the world model-based AORPO method fluctuating around -100, while that of PR2 fluctuating significantly around -203.2. This indicates that these methods have not accurately captured the opponent's behavioral patterns, leading the agents to make decisions based on erroneous predictions, resulting in a disadvantaged position. Furthermore, establishing environmental dynamics by AORPO based on MBPO's world model proves to be challenging.

**Comparison of different reasoning abilities.** Both parties always learn policies independently, but their reasoning abilities differ depending on whether the blue team independently learns an H2IL-MBOM to reason about the opponent's mental states or adopts a historical version of the red team's H2IL-MBOM. The results of Figure 2d indicate that the independent learning of an H2IL-MBOM by the blue team still achieves comparable performance over time.

## 4.2 Analysis of Opponents' Multiple Intentions and Latent Strategies

We study the cumulative error in predicting opponent intentions and strategies of two opposing teams, as well as the t-SNE (Van der Maaten & Hinton (2008)) distribution of these mental state representations for each agent over three time periods per episode ($steps <= 500, 1500 < steps <= 2500$, and $5000 < steps <= 6000$). Figures 3a and 3b show that both teams can quickly infer the opponent's mental states and have discovered an interesting phenomenon: after the reward curve converges, both teams suddenly grasp the key features or patterns at 6, 8.1, 11, and 12.5M steps, leading to a continuous decrease in cumulative prediction error of the opponent's mental states. This is mainly due to two factors. 1) Model complexity: in the initial stages of learning, the model gets stuck in local minima due to high complexity. However, as training progresses, the model gradually optimizes; 2) Data distribution changes: as the agent's strategy converges, the variation in the environment state space it explores becomes smaller, prompting the model to better capture current features. This indicates that strategy and model mutually reinforce each other, making predictions of the opponent's mental states more accurate.

The visualized t-SNE, as shown in Figures 3c and 3d, also present interesting phenomenons: using the predictions of agent 0 as an example, the intentions of opponents predicted exhibit multiple continuous strip distributions across three stages, rather than discrete clusters while the distribution of multiple predicted strategies is separable in each period time. These suggest that H2IL can capture the features of opponents' mental states. Specifically, within multiple smaller time intervals, there is a certain sequential relationship maintained among the opponent's mental states after dimension reduction, demonstrating the coherence of opponents' mental states. Simultaneously, multiple distributions correspond to stages of change of opponents' intentions and latent strategies (such as nose-to-nose approach, tailing, evasion, and missile launch), indicating the diversity and continuous transitions of opponents' strategies or tactics. We can also see that the number of intention transitions for all opponents in different stages is 11, 7, and 3, respectively, while the opponents' low-level strategies performs more smoothly and continuously. The feature distribution of opponents' intentions and strategies predicted by all agents and corresponding visualized maneuver trajectories are detailed in Appendix A.13, in which the average number (3,2,1) of changes in the opponent's intention predicted by each UAV in the three stages is consistent with the number of changes in the opponent's actual intention. In short, it is evident that our method not only allows for a global prediction of opponents' intentions but also a more precise capture of different behavior patterns and the evolving regularity of each opponent's latent strategies across three time periods, enhancing the interpretability.

## 4.3 Ablation Study

In this ablation study, we study the importance of each module in H2IL-MBOM by removing the low-level world model related to latent strategies (only intentions inference version), all history encoders (vanilla world model), Transformer and HEA, transition model, replacing GTr with local time Transformer, and replacing hypernetwork-add operator with share network-add operator in the Transformer.

As shown in Figure 4a, when only considering the opponent's intention, the learning performance will show a significant decline in the middle stage. This result indicates the importance of inferring

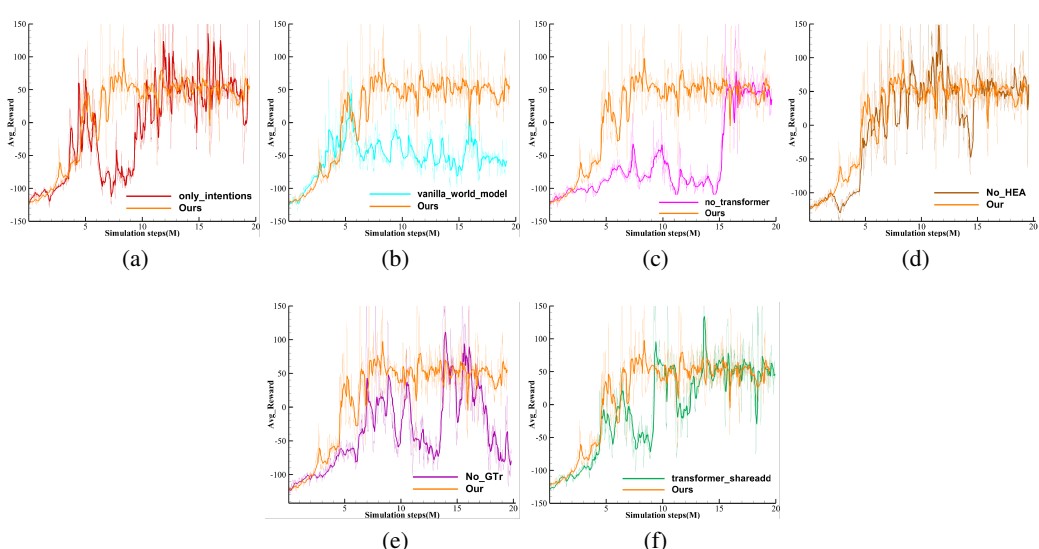

Figure 4: Results for ablation study.

low-level strategies for opponents and emphasizes the role of low-level world models. Furthermore, when only using the current observation for inference, as shown in Figure 4b, the reward falls into a local optimum. Because the opponent is also learning the opponent's model and their own strategy, the historical trajectory and strategy are dynamically changing. When the agent undergoes short-sighted training, it cannot obtain long-term reward returns. Consequently, using a vanilla world model for inference in highly dynamic environments often results in the agents' strategies becoming trapped in suboptimal solutions. This highlights the importance of utilizing historical encoders to infer the opponent's intentions and strategies, as they allow for quick adaptation to the opponents evolving and transitioning behaviors. Figures 4c-4f mainly verify the influence of transformer, HEA, GTr, and hyper operator on the design modules H2TE-MITD and LHTE-MLTD. Experiments have shown that when these modules are removed, there is a significant decrease in the convergence speed and stability of the reward curve. Therefore, it proves the importance of designing modules. In summary, the ablation experiments demonstrate the importance of our designed module and support the conclusions.

## 5    CONCLUSIONS

This paper proposes a novel opponent modeling method that integrates multiple intentions and latent strategies inference into the world model. We use a hierarchical architecture to study the impact and importance of opponents' intentions on their latent strategies and to predict the trajectories of teammates and opponents. Additionally, we propose MSORA-PPO, which allows both teams to learn their own H2IL-MBOM, infer adversarial strategies and intentions based on their own historical observations, and integrate the opponents' mental states inferred by H2IL-MBOM with local observations to independently learn policies and make decisions. This enables both teams to quickly capture and adapt to changes in the intentions and strategies of multiple opponents, as well as address the non-stationarity issues brought about by their continuous learning.

REPRODUCIBILITY STATEMENT

We have provided detailed designs of transition model, HDIRF, and LDLRF in the Appendix A.4,A.5, and A.6, respectively. The training details including environmental settings, hyperparameters are shown in the AppendixA.2, 1, and A.11. Lastly, we provide the source code in the supplementary materials.

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

# A APPENDIX

## A.1 LIMITATIONS

This study still has some limitations. First, we did not integrate multi-source information, which is important in practice and requires more representation learning. Second, although we analyzed the t-SNE distribution of opponent intentions and strategies, we have not yet studied the driving factors behind these distributions, necessitating further techniques to analyze opponents in detail. Last but not least, the method has not been validated on physical devices.

## A.2 EXPERIMENT DETAILS

**Observation Space.** Each agent's observation space includes the ego-state $O_e$, observations relative to cooperative adjacent agents $O_c$, observations relative to opponents and encountered missiles $O_{opp}$. Concretely, $O_e$ comprises ego altitude, sine and cosine values of ego roll angle, sine and cosine values of ego pitch angle, and three velocity components in the body coordinate system; the observation relative to each neighbor includes three components $\{\Delta x_{i,j,t}, \Delta y_{i,j,t}, \Delta z_{i,j,t}\}_{j=1,\ldots,2m}$ of relative position and three components $\{\Delta V x_{i,j,t}, \Delta V y_{i,j,t}, \Delta V z_{i,j,t}\}_{j=1,\ldots,2m}$ of relative velocity in the northeast celestial coordinate system; in addition to the above information, $O_{opp}$ also includes antenna angle $\{ATA_{i,j,t}\}_{j=1,\ldots,2m}$, aspect angle $\{AA_{i,j,t}\}_{j=1,\ldots,2m}$, elevation angle $\{EA_{i,j,t}\}_{j=1,\ldots,2m}$, horizontal angle $\{HA_{i,j,t}\}_{j=1,\ldots,2m}$, and distance $\{\Delta D_{i,j,t}\}_{j=1,\ldots,2m}$ relative to each opponent and missile.

**Action Space.** Each F14 aircraft in an air-combat scenario has five continuous actions, including aileron angle, elevator angle, rudder angle, thrust, and sign of launching missiles. A sign value greater than 0 indicates that it can be launched, otherwise it will not be launched. The specific launch also depends on the attack angle, distance, and enemy survival number on the battlefield.

**Rewards.** Rewards primarily consist of distance-angle reward relative to opponents, height-angle reward relative to opponents, speed-angle reward relative to opponents, penalties for collisions (-5) and proximity between teammates, altitude safety reward, attack angle reward, crash penalties(-100), penalties for the number of missiles (-10), penalties for being killed (-100), rewards for killing opponents (+100), and survival rewards(+1).

## A.3 HYPERHD2TSSM

In the RSSM, hidden states are sequentially derived to accommodate sequential learning. By contrast, the TSSM deviates from this processing by concurrently computing each hidden state through the utilization of past states and actions, thereby facilitating parallelized training. It is important to acknowledge, however, that as the temporal extent (T) expands, so too does the volume of requisite historical information, consequently escalating the computational demands. In our transition model design, we posit that the historical joint latent state-action of the $n$ adjacent agents is crucial, so we utilize a hypernetwork to interactively generate latent weights across agents based on the estimated state from the last step and further predict the state change at the next step. With reasoning, the latent weights at each step implicitly contain the historical information about neighbors from the beginning of reasoning to the desired time, leading to the $O(1)$ complexity. The comparison with RSSM, TSSM, and HyperHD2TSSM can be found as follows: $z_t \sim q(z_t|h_t, O_t)$

Here, we utilize $HJLGT_I$, $HJLGT_L$, and a Gaussian model to approximate $p(z_{I,i,t}|z_{I,i,t-1}, z_{I,n_i,t-1}, a_{i,t-1}, a_{n_i,t-1})$ and $(z_{L,i,t}|z_{L,i,t-1}, z_{L,n_i,t-1}, a_{i,t-1}, a_{n_i,t-1}, z_{I,i,t})$. Within this framework, $w_{I,i,t}$, $w_{I,n_i,t}$ are the neural network weights for estimating of estimated intentions toward opponents generated by the hypernetwork for each agent and their corresponding neighbors. Similarly, $w_{L,i,t}$, $w_{L,n_i,t}$ are the neural network weights of for estimating latent strategies toward opponents for each agent and their corresponding neighbors. All agents within the same team share a common hierarchical world model. Through a hypernetwork, they can construct transition models HJLGT for each agent without increasing neural network parameters. This eliminates the need to for building individual decentralized world models for each agent, which is different from the centralized, shared, and decentralized world models, offering advantages akin to the of the latter two.

**Algorithm 1** MSOAR-PPO.

**Require:** $\leq step_{\max}$, total numbers $N$, observable numbers $n$, and missile numbers $n$ of read team agents,and total numbers $M$, observable numbers $m$, and missile numbers $m$ of blue team agents;
Initialize the network parameters of H2IL-MBOM of two teams: $\{\phi_I, \psi_I, \theta_I,, \theta_r, \phi_L, \psi_L, \theta_L\}$, and $\{\phi_I, \psi_I, \theta_I, \theta_r, \phi_L, \psi_L, \theta_L\}$, actor policies of two teams: $\pi_\theta$ and $\pi_\theta$, critic networks of two teams: $V_\psi$ and $V_\psi$;
Initialize the opponents' intentions $\{z_{I,i}\}_{i=1}^N$ reasoned by red team, and opponents' intentions $\{z_{I,j}\}_{j=1}^M$ reasoned by blue team;
Set learning rate $\alpha_{rl}$ of RL for red team and the learning rate $\alpha_m$ of their H2IL-MBOM, and learning rates $\alpha_{rl}, \alpha_m$ of blue team;
Initialize memory buffers $\{D_{env,t}\}_{t=1}^T$, $\{D_{env,t}\}_{t=1}^T$ and historical buffers $\{H_{opp,t}\}_{t=1}^{512}$, $\{H_{c,t}\}_{t=1}^{512}$, $\{H_{opp,t}\}_{t=1}^{512}$, $\{H_{c,t}\}_{t=1}^{512}$;
**while** $step \leq step_{\max}$ **do**
    Reinitialize the environment;
  **while** not done **do**
    **for** red team agents $i = 1, ..., N$ **do**
      Obtain the current observations $O_{opp,i,t} = \{O_{i,j,t}\}_{j=1}^{2m}$ and $O_{c,i,t} = \{O_{i,l,t}\}_{l=1}^n$ of each agent, and gather historical observations $H_{opp,t}$ and $H_{c,t}$;
      Infer opponents' intentions $\{z_{I,i,j,t}\}_{j=1}^{2m}$ with $q(z_{I,i,t}|H_{opp,t}, O_{opp,i,t})$ by eq.equation 3-equation 7;
      Infer opponents' latent strategies $\{z_{L,i,j,t}\}_{j=1}^{2m}$ with $q(z_{L,i,t}|H_{c,t}, O_{c,i,t}, z_{I,i,t})$ by eq.equation 8-equation 10;
      Select actions according to the policy $\pi_\theta(\cdot|O_{opp,i,t}, O_{c,i,t}, z_{I,i,t}, z_{L,i,t})$ with HEA;
    **end for**
    **for** blue team agents $j = 1, ..., M$ **do**
      Obtain the current observations $O_{opp,j,t} = \{O_{j,i,t}\}_{i=1}^{2n}$ and $O_{c,j,t} = \{O_{j,l,t}\}_{l=1}^m$ of each agent, and gather historical observations $H_{opp,t}$ and $H_{c,t}$;
      Infer opponents' intentions $\{z_{I,j,i,t}\}_{i=1}^{2n}$ with $q(z_{I,j,t}|H_{opp,t}, O_{opp,j,t})$ by eq.equation 3-equation 7;
      Infer opponents' latent strategies $\{z_{L,j,i,t}\}_{i=1}^{2n}$ with $q(z_{L,j,t}|H_{c,t}, O_{c,j,t}, z_{I,j,t})$ by eq.equation 8-equation 10;
      Select actions according to the policy $\pi_\theta(\cdot|O_{opp,j,t}, O_{c,j,t}, z_{I,j,t}, z_{L,j,t})$ with HEA;
    **end for**
    Execution actions, and obtain rewards and next states;
    Add transitions to $D_{env} \leftarrow D_{env} \cup (O_{i,t}, a_{i,t}, r_{i,t}, O_{i,t+1}, z_{I,i,t}, z_{L,i,t})$ and $D_{env} \leftarrow D_{env} \cup (O_{i,t}, a_{i,t}, r_{i,t}, O_{i,t+1}, z_{I,i,t}, z_{L,i,t})$;
  **end while**
  Train H2IL-MBOM of both teams by eq.1;
  **for** $k = 1$ to num-epoch **do**
    **// Update policy and critic of both teams by PPO, respectively**:
    Computer loss $J_\pi, J_c$ and $J_\pi, J_c$ of both teams from PPO;
    $\theta \leftarrow \theta + \alpha_{rl} \nabla_\theta J_\pi(O_t, z_{I,t}, z_{L,t})$;
    $\psi \leftarrow \psi - \alpha_{rl} \nabla_\psi J_c(O_t, z_{I,t}, z_{L,t})$;
    $\theta \leftarrow \theta + \alpha_{rl} \nabla_\theta J_\pi(O_t, z_{I,t}, z_{L,t})$;
    $\psi \leftarrow \psi - \alpha_{rl} \nabla_\psi J_c(O_t, z_{I,t}, z_{L,t})$;
  **end for**
  Clear up the respective memories;
**end while**

Table 1: Comparison of RSSM, TSSM, and HyperHD2TSSM

| | Rssm | Tssm | HyperHD2TSSM |
|---|---|---|---|
| Representation model | $z_t \sim q(z_t\|h_t, O_t)$ | $z_t \sim q(z_t\|O_t)$ | $z_{I,i,t} \sim q(z_{I,i,t}\|H_{opp,t}, O_{opp,i,t}),$ $z_{L,i,t} \sim q(z_{L,i,t}\|H_{c,t}, O_{c,i,t}, z_{I,i,t})$ |
| Deterministic model | $h_{t+1} = gru(h_t, z_t, a_t)$ | $h_{t+1} = Transformer(z_{1:t}, a_{1:t})$ | $w_{I,i,t+1}, w_{I,n_i,t+1} =$ $Hyper(z_{I,i,t}, a_{i,t}, z_{I,n_i,t}, a_{n_i,t}),$ $h_{I,i,t+1} =$ $HJLGT_{w_{I,n_i,t+1}}(z_{I,i,t}, z_{I,n_i,t}, a_{i,t}, a_{n_i,t})$ $w_{L,i,t+1}, w_{L,n_i,t+1} =$ $Hyper(z_{L,i,t}, a_{i,t}, z_{L,n_i,t}, a_{n_i,t}),$ $h_{L,i,t+1} =$ $HJLGT_{w_{L,n_i,t+1}}(z_{L,i,t}, z_{L,n_i,t}, a_{i,t}, a_{n_i,t}, z_{I,i,t})$ |
| Stochastic model | $\hat{z}_{t+1} \sim p(\hat{z}_{t+1}\|h_{t+1})$ | | $\Delta\hat{z}_{I,i,t+1} \sim p(\Delta\hat{z}_{I,i,t+1}\|h_{I,i,t+1}),$ $\hat{z}_{I,i,t+1} = \Delta\hat{z}_{I,i,t+1} + z_{I,i,t}$ $\Delta\hat{z}_{L,i,t+1} \sim p(\Delta\hat{z}_{L,i,t+1}\|h_{L,i,t}),$ $\hat{z}_{L,i,t+1} = \Delta\hat{z}_{I,i,t+1} + z_{L,i,t}$ |
| Observation model | $p(O_{t+1}\|z_{t+1}, h_{t+1})$ | | $p(O_{opp,i,t+1}\|z_{I,i,t+1}, h_{I,i,t+1}),$ $p(O_{c,i,t+1}\|z_{L,i,t+1}, h_{L,i,t+1})$ |
| Reward model | $p(r_{t+1}\|z_{t+1}, h_{t+1})$ | | $p(r_{i,t+1}\|z_{I,i,t+1}, h_{I,i,t+1}, z_{L,i,t+1}, h_{L,i,t+1})$ |

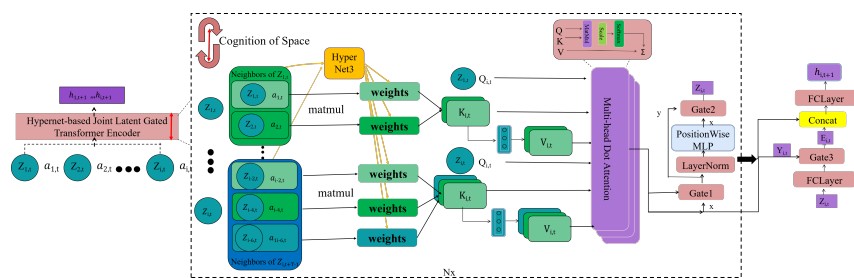

Figure 5: Architecture of the HJLGT.

## A.4 HJLGT

As shown in Figure 5, the HJLGT is defined as follow:

$$
\begin{aligned}
&i.e., h_{i,t+1} \leftarrow w_{i,t+1}, w_{n_i,t+1} \leftarrow z_{i,t}, z_{n_i,t}, a_{i,t}, a_{n_i,t} \\
&h_{i,t+1} = HJLGT_{w_{n_i,t}}(z_{i,t}, z_{n_i,t}, a_{i,t}, a_{n_i,t}): \\
&z_{i+n_i,t} = hstack(z_{i,t}, z_{n_i,t}) \\
&w_{i,t+1}, w_{n_i,t+1} = Hyper(z_{i,t}, a_{i,t}, z_{n_i,t}, a_{n_i,t}) \\
&w_{i+n_i,t+1} = hstack(w_{i,t+1}, w_{n_i,t+1}) \\
&Q_{i,t} = z_{i+n_i,t}, K_{i,t} = \text{Tanh}(z_{i+n_i,t}@w_{i+n_i,t+1}), V_{i,t} = K_{i,t}W_i^V \\
&x = MHA(Q_{i,t}, K_{i,t}, V_{i,t}) \\
&y = Gate1(x, x) \\
&z_{i,t} = Gate2(y, PositionWiseMlp(LayerNorm(y))) \\
&E_{i,t} = Gate3(x, FCLayer(z_{i,t})) \\
&h_{i,t+1} = FCLayer(Concat(E_{i,t}, x))
\end{aligned}
\tag{2}
$$

where the hstack operation involves stacking elements in a horizontal manner, MHA is the multi-head attention. It can be seen that the proposed transition model is designed for interactive prediction rather than independent prediction in a multi-agent system and can adaptively establish transition models for each agent without increasing model parameters, which makes it more adaptable and scalable.

## A.5 HIGH-LEVEL DYNAMIC INTENT-AWARE REPRESENTATION FUSION (HDIRF)

During each learning stage, historical states in the most recent steps undergo dynamic change. The intention queries within each MITD layer are derived from the outputs of the previous layer, adapting as the dynamics evolve. Consequently, by leveraging by H2TE and MITD, we can fuse historical observations specific to each agent. This fusion process enables dynamical learning of multiple intents as shown in Figure 6.

**High-level History Transformer Encoder (H2TE).** The historical observations $H_{opp,t} \in \mathbb{R}^{N \times 512 \times D}$ pertaining to opponents of all cooperative agents are encoded by HEA:

$$w_{H,i,j,t} = Hyper(H_{j,t}), e^{i,j,t} = \text{Tanh}(H_{j,t}@w_{H,i,j,t}), e^{i,t} = \frac{1}{m}\sum_{j=1}^{m} e^{i,j,t},$$
$$\alpha^{i,j,t} = soft\max(MLP([repeat(e^{i,t}), e^{i,j,t}])), AttH_{i,t} = \frac{1}{m}\sum_{j=1}^{m}\alpha^{i,j,t}\varphi_h(e^{i,j,t}) \tag{3}$$

where $Hyper()$ operator is defined in A.7.1, 512 is the most recent step, $D = m \times d_m$ is the observation dimensionalities to $m$ opponents within the observation scope of each agent, $H_{j,t}$ is the observation history to $j - th$ the opponent of all agents, and $w_{H,i,j,t}$ is corresponding neural weights generated by hypernetwork, and all $AttH_{opp} \in \mathbb{R}^{N \times 512 \times C}$ capture space dependence at local time. Next, we employ Transformer architecture that adopts multi-head attention (MHA) in each layer to capture the global time dependence of $AttH'_{opp} = reshape(AttH_{opp}) \in \mathbb{R}^{512, N \times C}$, where each layer of H2TE operates as follows:

$$q = k = AttH'_{opp}, v = MLP(k), AH_{opp} = AttH'_{opp} + MHA(q, k, v),$$
$$AttH'_{opp} = LayerNorm(AH_{opp} + MLP(LayerNorm(AH_{opp}))) \tag{4}$$

Finally, we reshape the output $AttH'_{opp}$ to $AttH_{opp} \in \mathbb{R}^{N \times 512 \times C}$. This operation is referred to as $GTr()$. Due to the advantages of the HEA, the updated historical feature of opponents captures the interdependence between our own agents and corresponding opponent agents at the same time point in historical observations, and also benefits from the advantages of the GTr to capture the similar patterns of opponent agent behaviors at different time points, constructing a more macroscopic perspective on time series similarity and the development of opponent agent behavior.

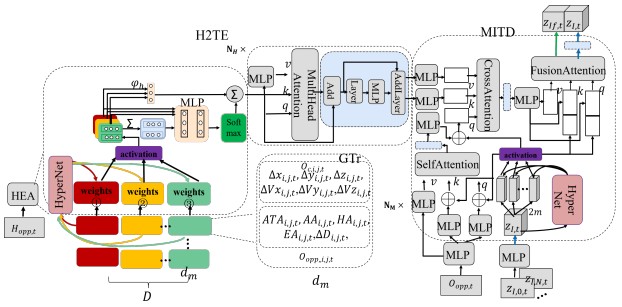

Figure 6: The structure of HDIRF that comprises H2TE and MITD. The HDIRF incorporates given observations regarding opponents and multi-learnable intention queries generated by a hypernetwork for interactive intention feature predictions.

**Multi Intention Transformer Decoder (MITD).** Given the historical feature of opponents $AttH_{opp}$ and current observation $O_{opp} \in \mathbb{R}^{T \times N \times 2m \times d_m}$ regarding opponents, the MITD is employed for multi-intention prediction of opponents and launched missiles, where $2m$ is the number of opponents and missiles launched. Due to the unknown opponent's intentions, we initialize the $2m$ dynamic intention queries $z_I \in \mathbb{R}^{T \times N \times 2m \times d_I}$, where $z_I$ evolve along with $O_{opp}$ over time, both in terms of content and quantity. To maintain simplicity, we have omitted the MHA, layer normalization, Feed-Forward network, and residual connections within following each module of the transformer layer.

**a) Hypernetwork-based Intention Self-attention Module.** This module fuses embedding $O_{opp,e} = MLP(O_{opp})$ of $O_{opp}$ and $z_I$ to propagate information among $2m$ dynamic intentions. Each intention query is encoded by the hypernetwork and is added to the embedding of current observation content regarding the opponent:

$$w_{I,i,j,t} = Hyper(z_{I,i,j,t}), q_{Ih,i,j,t} = \text{Tanh}(z_{I,i,j,t}@w_{I,i,j,t}), q_{Ih} = \{q_{Ih,i,j,t}\}_{i=1,...N;j=1,...,2m}^{t=t_0,...,t_0+T},$$
$$q_{I,s} = MLP(O_{opp,e}) + q_{Ih}, k_{I,s} = MLP(O_{opp,e}) + q_{Ih}, v_{I,s} = MLP(O_{opp,e}) \tag{5}$$

where $w_{I,i,j,t}$ and $z_{I,i,j,t}$ are assigned neural network weight and initialized intention query of each opponent of each agent at each time, i.e., $z_I = \{z_{I,i,j,t}\}_{i=1,...N;j=1,...,2m}^{t=t_0,...,t_0+T}$, and $q_{Ih}$ are the projection

of initialized multi-intention queries; therefore, the intention feature queries $q_{I,s} \in \mathbb{R}^{T \times N \times 2m \times C}$ in the self-attention module are updated.

**b) Hypernetwork-based Intention Cross-attention Module.** The module queries each intention prediction of each agent at the assembled historical feature of opponents of all cooperative agents. Considering the output $q_{I,s}$ of self-attention and initial intention query $q_{Ih}$ as intention queries for cross-attention, and the output of the H2TE module as intention key values, the updated intention feature queries $q_{I,c} \in \mathbb{R}^{T \times (2Nm) \times C}$ in the cross-attention module are obtained:

$$q_{I,c} = MLP(q_{I,s}) + q_{Ih}, k_{I,c} = MLP(AttH_{opp}), v_{I,c} = MLP(AttH_{opp}), q_{I,c} = reshape(q_{I,c}),$$
$$\in \mathbb{R}^{T \times (2Nm) \times C}, k_{I,c} = reshape(k_{I,c}) \in \mathbb{R}^{1 \times (512N) \times C}, v_{I,c} = reshape(v_{I,c}) \in \mathbb{R}^{1 \times (512N) \times C} \quad (6)$$

**c) Hypernetwork-based Intention Fusion Module.** In scenarios where all cooperative agents face the same opponent, it is crucial to infer which allied agent the opponent's intention will threaten. Hence, we incorporate the fusion module to capture and synthesize the intricate interactions of each intention prediction across agents.

$$q_{I,c} = reshape(q_{I,c}) \in \mathbb{R}^{T \times 2m \times N \times C}, q_{Ih} = reshape(q_{Ih}) \in \mathbb{R}^{T \times 2m \times N \times C}, q_{I,f} = [MLP(q_{I,c}), q_{Ih}],$$
$$k_{I,f} = [MLP(q_{I,c}), q_{Ih}], v_{I,f} = MLP(q_{I,c}) \quad (7)$$

where $q_{I,c}$ are the outputs of intention cross-attention module, $q_{I,c}$ and $q_{Ih}$ are aggregated by concatenation to obtain new feature, and $q_{I,f} \in \mathbb{R}^{T \times 2m \times N \times 2C}, k_{I,f} \in \mathbb{R}^{T \times 2m \times N \times 2C}, v_{I,f} \in \mathbb{R}^{T \times 2m \times N \times C}$. Finally, $z_I$ are the updated intent feature prediction in each layer of MITD.

## A.6 Low-level Dynamic Latent-strategy-aware Representation Fusion (LDLRF)

After obtaining inferred intention, the next step is to further infer latent strategies and understand how latent strategies react to intent prediction. Therefore, we fuse the historical observations $H_{c,t} \in \mathbb{R}^{N \times 512 \times D}$ to cooperative agents, current observations $O_c$ to cooperative neighbors, and intent prediction by LHTE and MLTD to dynamically learn multiple latent strategies. In the LDLRF layer, as shown in Figure 7, the multi-dynamic latent strategies are initialized by the intentions feature $z_I$ generated by MITD: $z_L = MLP([Gate(z_{If}, MLP(z_I)), z_{If}]) \in \mathbb{R}^{T \times 2m \times N \times C}$, and the embedding of current observation to cooperative neighbors $E_c = HEA(O_c) \in \mathbb{R}^{T \times N \times C}$. $Gate$ operator is defined in Appendix A.7.2.

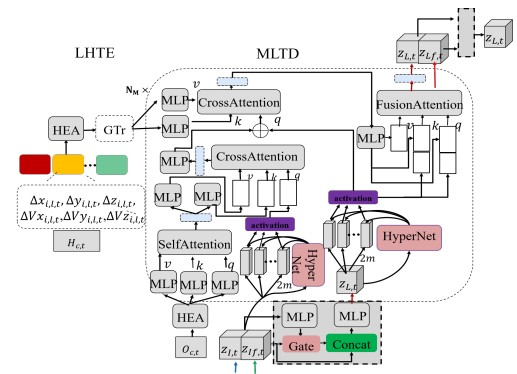

Figure 7: The structure of LDLRF that comprises LHTE and MLTD. The LDLRF incorporates observations of cooperative neighbors and latent strategy queries initialized by intention queries to capture the dynamic impact of multiple intentions on strategy decisions.

**Low-level History Transformer Encoder (LHTE).** At this level, $H_{c,t}$ are encoded by the same but separate HEA and GTr: $AttH_c = HEA(H_{c,t}) \in \mathbb{R}^{N \times 512 \times C}$, $AttH_c = reshape(GTr(reshape(AttH_c)))$.

**Multi Latent Strategy Transformer Decoder (MLTD).** The MLTD is composed of three modules, as follows:

**a) Hypernetwork-based Intention Self-Cross-attention Module.** To capture the influence of the opponents' intentions, we first conduct self-attention by $E_c$, and then apply cross-attention to each specific intention $z_{I,i,j,t}$.

$$
\begin{aligned}
&q_{Ls} = MLP(E_c), k_{Ls} = MLP(E_c), v_{Ls} = MLP(E_c), \\
&k_{LI} = LayerNorm(E_c + MHA(q_{Ls}, k_{Ls}, v_{Ls})), k_{LI} = MLP(k_{LI}), v_{LI} = MLP(k_{LI}), \\
&w_{L,i,j,t} = Hyper(z_{I,i,j,t}), q_{LI,i,j,t} = \text{Tanh}(z_{I,i,j,t}@w_{L,i,j,t}), q_{LI} = \{q_{LI,i,j,t}\}_{i=1,\ldots N;j=1,\ldots,2m}^{t=t_0,\ldots,t_0+T}, \\
&q_{LI} = reshape(q_{LI}) \in \mathbb{R}^{T\times(2Nm)\times C}, k_{LI}, v_{LI} \in \mathbb{R}^{T\times N\times C}
\end{aligned}
\tag{8}
$$

**b) Hypernetwork-based Latent-strategy Cross-attention Module.** Similarly, we aggregate latent-strategy queries by the outputs $q_{L,sc}$ of the previous module and apply the cross-attention to historical observations regarding cooperative agents and each specific latent strategy $z_{L,i,j,t}$.

$$
\begin{aligned}
&q_{Lc} = MLP(q_{L,sc}), w_{L,i,j,t} = Hyper(z_{L,i,j,t}), q_{Lh,i,j,t} = \text{Tanh}(z_{L,i,j,t}@w_{L,i,j,t}), \\
&q_{Lh} = \{q_{Lh,i,j,t}\}_{i=1,\ldots N;j=1,\ldots,2m}^{t=t_0,\ldots,t_0+T}, q_{Lh} = reshape(q_{Lh}) \in \mathbb{R}^{T\times(2Nm)\times C}, q_{L,c} = q_{Lc} + q_{Lh}, \\
&k_{L,c} = reshape(MLP(AttH_c)) \in \mathbb{R}^{1\times(512N)\times C}, v_{L,c} = reshape(MLP(AttH_c)) \in \mathbb{R}^{1\times(512N)\times C}
\end{aligned}
\tag{9}
$$

**c) Hypernetwork-based Latent-strategy Fusion Module.** Finally, the Latent-strategy Fusion Module is introduced to capture interactions of latent strategy across agents.

$$
\begin{aligned}
&q_{L,c} = reshape(q_{L,c}) \in \mathbb{R}^{T\times 2m\times N\times C}, q_{Lh} = reshape(q_{Lh}) \in \mathbb{R}^{T\times 2m\times N\times C}, q_{L,f} = [MLP(q_{L,c}), q_{Lh}], \\
&k_{L,f} = [MLP(q_{L,c}), q_{Lh}], v_{L,f} = MLP(q_{L,c})
\end{aligned}
\tag{10}
$$

where $q_{L,c}$ are the outputs of the latent-strategy cross-attention module, $q_{L,c}$ and $q_{Lh}$ are aggregated by concatenation to obtain new features, and the updated latent strategies $z_L \in \mathbb{R}^{T\times 2m\times N\times C}$ of opponents under the affection of $z_I$ are obtained in each layer of MLTD.

## A.7 THE DEFINITIONS OF OPERATORS

### A.7.1 HYPER OPERATOR

The Hyper operator is defined as follow:

$$
\begin{aligned}
&x = za_{i,t} = Concat(z_{i,t}, a_{i,t}); \\
&w_{i,t} = HyperNet(x; \theta_{hyper}); \\
&y = f(x; w_{i,t}) = f(x; HyperNet(x; \theta_{hyper}));
\end{aligned}
\tag{11}
$$

where we assume that the dimensions of concatenation $za_{i,t}$ of $z_{i,t}$ and $a_{i,t}$ are $[n, d_z + d_a]$. Initially, the hypernetwork with $\theta_{hyper}$ is sized as $[d_z + d_a, (d_z + d_a) \times d_h]$, and it is multiplied by $za_{i,t}$ to produce weights of size $[n, (d_z + d_a) \times d_h]$. To automate weight assignment and create a reduced neural network, $za_{i,t}$ is reshaped to $[n, 1, d_z + d_a]$ using the unsqueeze operator and weights with the size of $[n, (d_z + d_a) \times d_h]$ is reshaped to $[n, d_z + d_a, d_h]$. Finally, we multiply and activate them using $Tanh$ function to obtain results while the the size of results is transformed into dimensions $[n, d_h]$. This process is denoted as $w_{i,t} = Hyper(z_{i,t}, a_{i,t})$

### A.7.2 GATE OPERATOR

The Gate operator is defined as follow:

$$
\begin{aligned}
&Gate(y, x) = (1 - z) \odot y + z \odot h; \\
&z = \sigma(W_z x + U_z y - b_g); \\
&h = \tanh(W_g x + U_g(r \odot y)); \\
&r = \sigma(W_r x + U_r y);
\end{aligned}
\tag{12}
$$

where $\odot$ is the hadamard product, which refers to the element-wise multiplication of two matrices of the same size; $\sigma$ is the sigmoid operation; the linear weights $W_z$, $U_z$, $W_g$, $U_g$, $W_r$, and $U_r$, along with the bias $b_g$, are components used in the model.

### A.8 Derivation of the Hierarchical Variational Lower Bound

The joint probability and the hierarchical evidence lower bound (HELBO) are derived as follows:

$$
\begin{aligned}
&p(O_{opp,1:N,1:T}, O_{c,1:N,1:T}, a_{1:N,1:T}, h_{I,1:N,1:T}, z_{I,1:N,1:T}, h_{L,1:N,1:T}, z_{L,1:N,1:T}) \\
&= \prod_{t=1}^{T} \left[
\begin{array}{l}
p(h_{I,1:N,t}, z_{I,1:N,t} | z_{I,1:N,t-1}, a_{1:N,t-1}) p(O_{opp,1:N,t} | h_{I,1:N,t}, z_{I,1:N,t}) \\
p(h_{L,1:N,t}, z_{L,1:N,t} | z_{L,1:N,t-1}, a_{1:N,t-1}, z_{I,1:N,t}) p(O_{c,1:N,t} | h_{L,1:N,t}, z_{L,1:N,t}) \\
p(a_{1:N,t} | O_{opp,1:N,t}, O_{c,1:N,t}, z_{I,1:N,t}, z_{L,1:N,t})
\end{array}
\right] \\
&= \prod_{t=1}^{T} \left[
\begin{array}{l}
p(z_{I,1:N,t} | h_{I,1:N,t}) p(h_{I,1:N,t} | z_{I,1:N,t-1}, a_{1:N,t-1}) p(O_{opp,1:N,t} | h_{I,1:N,t}, z_{I,1:N,t}) \\
p(z_{L,1:N,t} | h_{L,1:N,t}) p(h_{L,1:N,t} | z_{L,1:N,t-1}, a_{1:N,t-1}, z_{I,1:N,t}) p(O_{c,1:N,t} | h_{L,1:N,t}, z_{L,1:N,t}) \\
p(a_{1:N,t} | O_{opp,1:N,t}, O_{c,1:N,t}, z_{I,1:N,t}, z_{L,1:N,t})
\end{array}
\right] \\
&= \prod_{t=1}^{T} \left[
\begin{array}{l}
p(z_{I,1:N,t} | z_{I,1:N,t-1}, a_{1:N,t-1}) p(O_{opp,1:N,t} | h_{I,1:N,t}, z_{I,1:N,t}) \\
p(z_{L,1:N,t} | z_{L,1:N,t-1}, a_{1:N,t-1}, z_{I,1:N,t}) p(O_{c,1:N,t} | h_{L,1:N,t}, z_{L,1:N,t}) \\
p(a_{1:N,t} | O_{opp,1:N,t}, O_{c,1:N,t}, z_{I,1:N,t}, z_{L,1:N,t})
\end{array}
\right] \\
&= \prod_{t=1}^{T} \left[
\begin{array}{l}
p(z_{I,1,t} | z_{I,1,t-1}, z_{I,n_1,t-1}, a_{1,t-1}, a_{1,n_1,t-1}) ... p(z_{I,N,t} | z_{I,N,t-1}, z_{I,n_N,t-1}, a_{N,t-1}, a_{N,n_N,t-1}) \\
p(O_{opp,1,t} | h_{I,1,t}, z_{I,1,t}) ... p(O_{opp,N,t} | h_{I,N,t}, z_{I,N,t}) \\
p(z_{L,1,t} | z_{L,1,t-1}, z_{L,n_1,t-1}, a_{1,t-1}, a_{1,n_1,t-1}, z_{I,1,t}) ... p(z_{L,N,t} | z_{L,N,t-1}, z_{L,n_N,t-1}, a_{N,t-1}, a_{N,n_N,t-1}, z_{I,N,t}) \\
p(O_{c,1,t} | h_{L,1,t}, z_{L,1,t}) ... p(O_{c,N,t} | h_{L,N,t}, z_{L,N,t}) \\
p(a_{1,t} | O_{opp,1,t}, O_{c,1,t}, z_{I,1,t}, z_{L,1,t}) ... p(a_{N,t} | O_{opp,N,t}, O_{c,N,t}, z_{I,N,t}, z_{L,N,t})
\end{array}
\right] \\
&= \prod_{t=1}^{T} \prod_{i=1}^{N} \left[
\begin{array}{l}
p(z_{I,i,t} | z_{I,i,t-1}, z_{I,n_i,t-1}, a_{i,t-1}, a_{n_i,t-1}) p(O_{opp,i,t} | h_{I,i,t}, z_{I,i,t}) \\
p(z_{L,i,t} | z_{L,i,t-1}, z_{L,n_i,t-1}, a_{i,t-1}, a_{n_i,t-1}, z_{I,i,t}) p(O_{c,i,t} | h_{L,i,t}, z_{L,i,t}) \\
p(a_{i,t} | O_{opp,i,t}, O_{c,i,t}, z_{I,i,t}, z_{L,i,t})
\end{array}
\right]
\end{aligned}
\tag{13}
$$

$$
\begin{aligned}
&\log p(O_{opp,1:N,1:T}, O_{c,1:N,1:T}, a_{1:N,1:T}, h_{I,1:N,1:T}, z_{I,1:N,1:T}, h_{L,1:N,1:T}, z_{L,1:N,1:T}) \\
&= \log E_{q(z_{1:N,1:T}|H_{1:T},O_{1:N,1:T})} \left[ \frac{p(O_{opp,1:N,1:T},O_{c,1:N,1:T},a_{1:N,1:T},h_{I,1:N,1:T},z_{I,1:N,1:T},h_{L,1:N,1:T},z_{L,1:N,1:T})}{q(z_{1:N,1:T}|H_{1:T},O_{1:N,1:T})} \right] \\
&\geq E_{q(z_{1:N,1:T}|H_{1:T},O_{1:N,1:T})} \log \left[ \frac{p(O_{opp,1:N,1:T},O_{c,1:N,1:T},a_{1:N,1:T},h_{I,1:N,1:T},z_{I,1:N,1:T},h_{L,1:N,1:T},z_{L,1:N,1:T})}{q(z_{1:N,1:T}|H_{1:T},O_{1:T})} \right] \\
&= \int q(z_{1:N,1:T}|H_{1:T},O_{1:N,1:T}) \log \left[ \frac{p(O_{opp,1:N,1:T},O_{c,1:N,1:T},a_{1:N,1:T},h_{I,1:N,1:T},z_{I,1:N,1:T},h_{L,1:N,1:T},z_{L,1:N,1:T})}{q(z_{1:N,1:T}|H_{1:T},O_{1:T})} \right] \\
&\quad dz_{1:N,1:T} \\
&\quad\quad q(z_{I,1:N,1:T}|H_{opp,1:T},O_{opp,1:N,1:T}) q(z_{L,1:N,1:T}|H_{c,1:T},O_{c,1:N,1:T},z_{I,1:N,1:T}) \\
&= \int \sum_{t=1}^{T} \log \left[
\frac{
\begin{array}{l}
p(z_{I,1:N,t}|z_{I,1:N,t-1},a_{1:N,t-1}) p(O_{opp,1:N,t}|h_{I,1:N,t},z_{I,1:N,t}) \\
p(z_{L,1:N,t}|z_{L,1:N,t-1},a_{1:N,t-1},z_{I,1:N,t}) p(O_{c,1:N,t}|h_{L,1:N,t},z_{L,1:N,t}) \\
p(a_{1:N,t}|O_{opp,1:N,t},O_{c,1:N,t},z_{I,1:N,t},z_{L,1:N,t})
\end{array}
}{q(z_{I,1:N,t}|H_{opp,t},O_{opp,1:N,t}) q(z_{L,1:N,t}|H_{c,t},O_{c,1:N,t},z_{I,1:N,t})}
\right] dz_{1:N,1:T}
\end{aligned}
$$

$$
= \sum_{t=1}^{T} \left\{
\begin{array}{l}
\int \begin{array}{l} q(z_{I,1:N,1:t}|H_{opp,1:t},O_{opp,1:N,1:t}) q(z_{L,1:N,1:t}|H_{c,1:t},O_{c,1:N,1:t},z_{I,1:N,1:t}) \\ \log[p(O_{opp,1:N,t}|h_{I,1:N,t},z_{I,1:N,t})] \end{array} dz_{I,1:N,1:t} \\
+ \int \begin{array}{l} q(z_{I,1:N,1:t}|H_{opp,1:t},O_{opp,1:N,1:t}) q(z_{L,1:N,1:t}|H_{c,1:t},O_{c,1:N,1:t},z_{I,1:N,1:t}) \\ \log[p(O_{c,1:N,t}|h_{L,1:N,t},z_{L,1:N,t})] \end{array} dz_{L,1:N,1:t} \\
+ \int \begin{array}{l} q(z_{I,1:N,1:t}|H_{opp,1:t},O_{opp,1:N,1:t}) q(z_{L,1:N,1:t}|H_{c,1:t},O_{c,1:N,1:t},z_{I,1:N,1:t}) \\ \log[p(a_{1:N,t}|O_{opp,1:N,t},O_{c,1:N,t},z_{I,1:N,t},z_{L,1:N,t})] \end{array} dz_{1:N,1:t} \\
+ \int \begin{array}{l} q(z_{I,1:N,1:t}|H_{opp,1:t},O_{opp,1:N,1:t}) q(z_{L,1:N,1:t}|H_{c,1:t},O_{c,1:N,1:t},z_{I,1:N,1:t}) \\ \log \left[ \frac{p(z_{I,1:N,t}|z_{I,1:N,t-1},a_{1:N,t-1})}{q(z_{I,1:N,t}|H_{opp,t},O_{opp,1:N,t})} \right] \end{array} dz_{I,1:N,1:t} \\
+ \int \begin{array}{l} q(z_{I,1:N,1:t}|H_{opp,1:t},O_{opp,1:N,1:t}) q(z_{L,1:N,1:t}|H_{c,1:t},O_{c,1:N,1:t},z_{I,1:N,1:t}) \\ \log \left[ \frac{p(z_{L,1:N,t}|z_{L,1:N,t-1},a_{1:N,t-1},z_{I,1:N,t})}{q(z_{L,1:N,t}|H_{c,t},O_{c,1:N,t},z_{I,1:N,t})} \right] \end{array} dz_{L,1:N,1:t}
\end{array}
\right\}
$$

$$
= \sum_{t=1}^{T} \left\{
\begin{array}{l}
\int q(z_{I,1:N,1:t}|H_{opp,1:t},O_{opp,1:N,1:t}) \log[p(O_{opp,1:N,t}|h_{I,1:N,t},z_{I,1:N,t})] dz_{I,1:N,1:t} \\
+ \int q(z_{L,1:N,1:t}|H_{c,1:t},O_{c,1:N,1:t},z_{I,1:N,1:t}) \log[p(O_{c,1:N,t}|h_{L,1:N,t},z_{L,1:N,t})] dz_{L,1:N,1:t} \\
+ \int \begin{array}{l} q(z_{I,1:N,1:t}|H_{opp,1:t},O_{opp,1:N,1:t}) q(z_{L,1:N,1:t}|H_{c,1:t},O_{c,1:N,1:t},z_{I,1:N,1:t}) \\ \log[p(a_{1:N,t}|O_{opp,1:N,t},O_{c,1:N,t},z_{I,1:N,t},z_{L,1:N,t})] \end{array} dz_{1:N,1:t} \\
+ \int q(z_{I,1:N,1:t}|H_{opp,1:t},O_{opp,1:N,1:t}) \log \left[ \frac{p(z_{I,1:N,t}|z_{I,1:N,t-1},a_{1:N,t-1})}{q(z_{I,1:N,t}|H_{opp,t},O_{opp,1:N,t})} \right] dz_{I,1:N,1:t} \\
+ \int q(z_{L,1:N,1:t}|H_{c,1:t},O_{c,1:N,1:t},z_{I,1:N,1:t}) \log \left[ \frac{p(z_{L,1:N,t}|z_{L,1:N,t-1},a_{1:N,t-1},z_{I,1:N,t})}{q(z_{L,1:N,t}|H_{c,t},O_{c,1:N,t},z_{I,1:N,t})} \right] dz_{L,1:N,1:t}
\end{array}
\right\}
$$

$$
\begin{aligned}
&\{\int q(z_{I,1,1:t}|H_{opp,1:t},O_{opp,1,1:t})...q(z_{I,N,1:t}|H_{opp,1:t},O_{opp,N,1:t})\log[p(O_{opp,1,t}|h_{I,1,t},z_{I,1,t})]... \\
&\quad p(O_{opp,N,t}|h_{I,N,t},z_{I,N,t})]dz_{I,1:N,1:t} \\
&+\int q(z_{L,1,1:t}|H_{c,1:t},O_{c,1,1:t},z_{I,1,1:t})...q(z_{L,N,1:t}|H_{c,1:t},O_{c,N,1:t},z_{I,N,1:t})\log[p(O_{c,1,t}|h_{L,1,t}, \\
&\quad z_{L,1,t})...p(O_{c,N,t}|h_{L,N,t},z_{L,N,t})]dz_{L,1:N,1:t} \\
&\quad q(z_{I,1,1:t}|H_{opp,1:t},O_{opp,1,1:t})q(z_{L,1,1:t}|H_{c,1:t},O_{c,1,1:t},z_{I,1,1:t})...q(z_{I,N,1:t}|H_{opp,1:t}, \\
&+\int O_{opp,N,1:t})q(z_{L,N,1:t}|H_{c,1:t},O_{c,N,1:t},z_{I,N,1:t})\log[p(a_{1,t}|O_{opp,1,t},O_{c,1,t},z_{I,1,t},z_{L,1,t})... \\
&\quad p(a_{N,t}|O_{opp,N,t},O_{c,N,t},z_{I,N,t},z_{L,N,t})]dz_{1:N,1:t}
\end{aligned}
$$

$$
= \sum_{t=1}^{T}
\begin{aligned}
&\quad q(z_{I,1,1:t}|H_{opp,1:t},O_{opp,1,1:t})...q(z_{I,N,1:t}|H_{opp,1:t},O_{opp,N,1:t}) \\
&+\int \log \left[\begin{array}{c} p(z_{I,1,t}|z_{I,1,t-1},z_{I,n_1,t-1},a_{1,t-1},a_{n_1,t-1})... \\ \frac{p(z_{I,N,t}|z_{I,N,t-1},z_{I,n_N,t-1},a_{N,t-1},a_{n_N,t-1})}{q(z_{I,1,t}|H_{opp,t},O_{opp,1,t})...q(z_{I,N,t}|H_{opp,t}O_{opp,N,t})} \end{array}\right] dz_{I,1:N,1:t} \\
&\quad q(z_{L,1,1:t}|H_{c,1:t},O_{c,1,1:t},z_{I,1,1:t})...q(z_{L,N,1:t}|H_{c,1:t},O_{c,N,1:t},z_{I,N,1:t}) \\
&+\int \log \left[\begin{array}{c} p(z_{L,1,t}|z_{L,1,t-1},z_{L,n_1,t-1},a_{1,t-1},a_{n_1,t-1},z_{I,1,t})... \\ \frac{p(z_{L,N,t}|z_{L,N,t-1},z_{L,n_N,t-1},a_{N,t-1},a_{n_N,t-1},z_{I,N,t})}{q(z_{L,1,t}|H_{c,t},O_{c,1,t},z_{I,1,t})...q(z_{L,N,t}|H_{c,t},O_{c,N,t},z_{I,N,t})} \end{array}\right] dz_{L,1:N,1:t} \quad \}
\end{aligned}
$$

$$
= \sum_{t=1}^{T}
\begin{aligned}
&\{\int \sum_{i=1}^{N} q(z_{I,i,1:t}|H_{opp,1:t},O_{opp,i,1:t})\log[p(O_{opp,i,t}|h_{I,i,t},z_{I,i,t})]dz_{I,i,1:t} \\
&+\int \sum_{i=1}^{N} q(z_{L,i,1:t}|H_{c,1:t},O_{c,i,1:t},z_{I,i,1:t})\log[p(O_{c,i,t}|h_{L,i,t},z_{L,i,t})]dz_{L,i,1:t} \\
&+\int \sum_{i=1}^{N} \begin{array}{c} q(z_{I,i,1:t}|H_{opp,1:t},O_{opp,i,1:t})q(z_{L,i,1:t}|H_{c,1:t},O_{c,i,1:t},z_{I,i,1:t})\log[p(a_{i,t}|O_{opp,i,t},O_{c,i,t},z_{I,i,t},z_{L,i,t})] \\ dz_{i,1:t} \end{array} \\
&+\int \sum_{i=1}^{N} q(z_{I,i,1:t}|H_{opp,1:t},O_{opp,i,1:t})\log \left[\frac{p(z_{I,i,t}|z_{I,i,t-1},z_{I,n_i,t-1},a_{i,t-1},a_{n_i,t-1})}{q(z_{I,i,t}|H_{opp,t},O_{opp,i,t})}\right] dz_{I,i,1:t} \\
&+\int \sum_{i=1}^{N} q(z_{L,i,1:t}|H_{c,1:t},O_{c,i,1:t},z_{I,i,1:t})\log \left[\frac{p(z_{L,i,t}|z_{L,i,t-1},z_{L,n_i,t-1},a_{i,t-1},a_{n_i,t-1},z_{I,i,t})}{q(z_{L,i,t}|H_{c,t},O_{c,i,t},z_{I,i,t})}\right] dz_{L,i,1:t}\}
\end{aligned}
$$

$$
= \sum_{t=1}^{T}\sum_{i=1}^{N}
\begin{aligned}
&E_{q(z_{I,i,1:t}|H_{opp,1:t},O_{opp,i,1:t})}(\log[p(O_{opp,i,t}|h_{I,i,t},z_{I,i,t})]) + E_{q(z_{L,i,1:t}|H_{c,i,1:t},O_{c,i,1:t},z_{I,i,1:t})} \\
&(\log[p(O_{c,i,t}|h_{L,i,t},z_{L,i,t})]) + E_{q(z_{I,i,1:t}|H_{opp,1:t},O_{opp,i,1:t})q(z_{L,i,1:t}|H_{c,1:t},O_{c,i,1:t},z_{I,i,1:t})} \\
&\log[p(a_{i,t}|O_{opp,i,t},O_{c,i,t},z_{I,i,t},z_{L,i,t})] - E_{q(z_{I,i,1:t}|H_{opp,1:t},O_{opp,i,1:t})}KL(q(z_{I,i,t}|H_{opp,t},O_{opp,i,t})|| \\
&p(z_{I,i,t}|z_{I,i,t-1},z_{I,n_i,t-1},a_{i,t-1},a_{n_i,t-1})) - E_{q(z_{L,i,1:t}|H_{c,1:t},O_{c,i,1:t},z_{I,i,1:t})} \\
&KL(q(z_{L,i,t}|H_{c,t},O_{c,i,t},z_{I,i,t})||p(z_{L,i,t}|z_{L,i,t-1},z_{L,n_i,t-1},a_{i,t},a_{n_i,t-1},z_{I,i,t}))
\end{aligned}
\tag{14}
$$

## A.9 Testing Results

The win rate (WR) and survival rate (SR) are as evaluation metrics. We first confront the opponents who adopt the baseline strategy that includes straight fly, rectangular trajectory maneuver and evasion of missiles, and pursuing the tail of our aircraft. The results show that our SR is the highest and achieves a $100\%$ WR in 4 vs. 4 scenarios as presented in Table 2. We then test the effectiveness of our method against our method and our method against MAPPO under different numbers of agents as presented in Table 3 and 4. We use SR to evaluate performance because a group with fewer agents may sacrifice less or equal to the other group. In most cases, both teams make equal sacrifices because of the same reasoning ability of both teams, and in a small number of cases (e.g., 4 vs. 6, 4 vs. 8, 6 vs. 8) where the quantity is at a disadvantage, the red team still destroys one more aircraft than the blue team. In situations where the red team has a numerical advantage, it can achieve $100\%$ superiority (e.g., 8 vs. 4, 10 vs. 4, 10 vs. 6). Additionally, the advantage ranges are further expanded when our method against MAPPO. (e.g., 4 vs.4, 6, 8,10; 6 vs. 4, 8,10; 8 vs. 4; 10 vs. 4, 10 vs. 6). The results also demonstrate our method is endowed with good generalization ability. Due to the fixed dimensions of other MARLs, it is not possible to complete adversarial tasks in different quantities. As shown in Figure 3(c), the relevant opponent modeling methods are unable to complete this task, so there is no adversarial testing with these methods.

Table 2: The results of our method against the baseline strategy in 4 vs. 4 scenarios.

| SR(WR) | Straight fly | Maneuver | Pursue |
|---|---|---|---|
| Ours | 4:1(100 %) | 4:2(100 %) | 4:0(100 %) |

Table 3: The results of the confrontation of different number agents of our method.

| SR (Ours vs. Ours) | 4 | 6 | 8 | 10 |
|---|---|---|---|---|
| 4 | 2:2 | 3:4 | 3:6 | 3:9 |
| 6 | 3:1 | 3:3 | 3:4 | 3:7 |
| 8 | 8:0 | 3:1 | 3:3 | 3:5 |
| 10 | 10:0 | 10:0 | 3:1 | 3:3 |

Table 4: The results of our method vs. MAPPO under different numbers of agents.

| SR (Ours vs. MAPPO) | 4 | 6 | 8 | 10 |
|---|---|---|---|---|
| 4 | 3:2 | 3:4 | 3:6 | 3:8 |
| 6 | 3:0 | 3:3 | 3:4 | 3:6 |
| 8 | 8:0 | 3:1 | 3:3 | 3:5 |
| 10 | 10:0 | 10:0 | 3:1 | 3:3 |

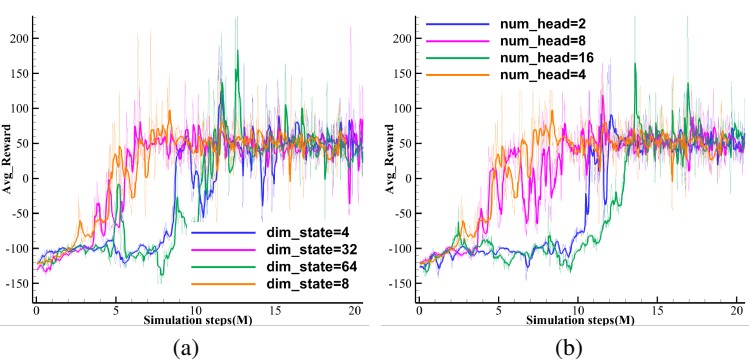

(a)                    (b)

Figure 8: The results on experiments with different hyperparameters: a) different dimensions of the mental states; b) different numbers of attention head

### A.10   Experiments about hyperparameters

We vary the dimensions of intentions from 4 to 64 and evaluate the impact of different dimensions on the performance of our method, as shown in Figure 8a. We observe that there is an optimal dimensions of intentions, 8, which maximizes the performance of the model. When the dimension of intentions is below 8 or above 32, it takes twice the time to converge, and the convergence speed is significantly reduced. Based on our experiments, the optimal number of attention heads is 8. At this optimal number, the model achieves the highest performance with lowest complexity.

Similarly, we vary the number of attention heads from 2 to 16 and measure the performance using the average rewards. As shown in Figure 1 8b, we observe that there is an optimal number of attention heads, 4, which maximizes the performance of the model. When the number of attention heads is below 4 or above 8, it also takes twice the time to converge, and the convergence speed is significantly reduced. Based on our experiments, the optimal number of attention heads is 4. At this optimal number, the model achieves the highest performance with lowest complexity.

In summary, the dimensions of the intention space and numbers of attention head are chosen based on the best balance between performance and computational efficiency.

### A.11   Hyperparameters

The hyperparameters are summarized in Table5 and Table6.

### A.12   Compute Resource

In our study, we performed simulations utilizing 36 parallel environments on a computer workstation equipped with dual Intel(R) Xeon(R) 40-core CPUs, 128 GB of RAM, and two NVIDIA RTX A4500 GPUs. Each environment completed 1500 maximum steps per episode at a simulation frequency of 60Hz. In total, there were roughly four days for training the air-combat environment.

Table 5: Hyperparameters of MARL

| Parameter | value |
|---|---|
| Interaction steps | $2 \times 10^7$ ($20M$) |
| Training steps | $1.58 \times 10^5$ |
| Learning rate | $3 \times 10^{-4}$ |
| Discount factor | 0.99 |
| Policy initialization | Xavier uniform |
| Optimizer | Adam |
| Gradient norm clipping | 5.0 |
| Rollout Length | 128 |
| Batch size | 1024 |
| Number of training epochs | 1 |
| Number of head | 4 |
| Attention size | 32 |
| Hidden state (models of policy and value) size | 128 |

Table 6: Hyperparameters of H2IL-MBOM

| Parameter | value |
|---|---|
| Training steps | $1.58 \times 10^5$ |
| Learning rate | $1 \times 10^{-4}$ |
| Discount factor | 0.99 |
| Optimizer | Adam |
| Gradient norm clipping | 5.0 |
| Number of head | 4 |
| Attention size | 32 |
| Intention $z_I$ and latent strategy $z_L$ dimensionality | 8 |
| Hidden state size | 32 |
| Number of layers $N_M$ and $N_H$ | 4 |

### A.13 VISUAL RESULTS

A shown in Figures 9 and 10, the visualization of scenarios depicting engagements between our method and MAPPO, as well as engagements between our method and itself, was conducted. The figures illustrate that during combat with MAPPO, our maneuver decisions were more agile and rapid, resulting in achieving a high altitude and angle advantage with a smaller flight radius, ultimately leading to a SR of 3:1. In confrontations with our own method, both sides exhibited similar reasoning capabilities, leading to primarily engaging in double loop motion, which represents a classic tactic in close-range aerial combat.

Combining Figures 9, 10,11, and 12, in the initial stage, the feature distribution range is relatively small, indicating both teams frequently make rapid maneuver transitions in a small space (such as climbing, making large turns to enter angles, and engaging in single-loop maneuvers). In the middle stage, both teams enter the engagement phase, conducting double-loop maneuvers (nose-to-nose approach and departure), and missile launches within a larger range. In the final stage, only alive agents engage in extensive pursuit and escape strategies. This is consistent with the average number of changes in the opponent's intention predicted by each UAV on average across three stages.

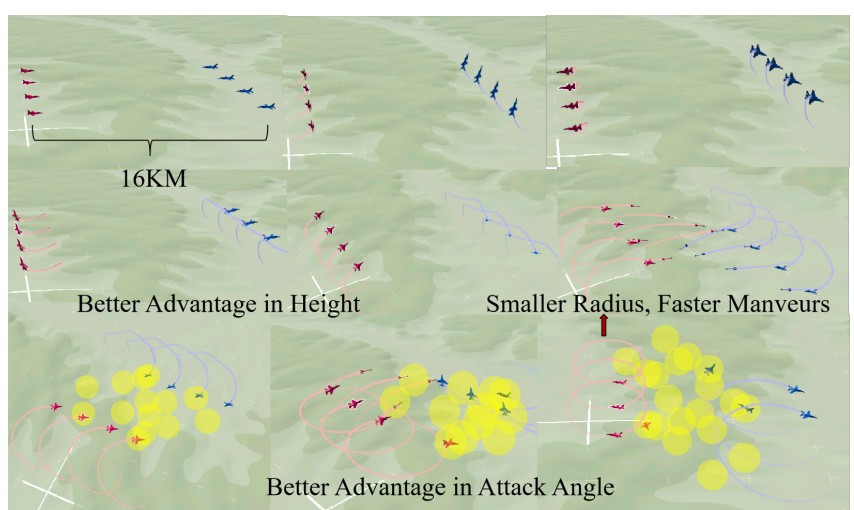

Figure 9: Snapshot of our method vs. MAPPO.

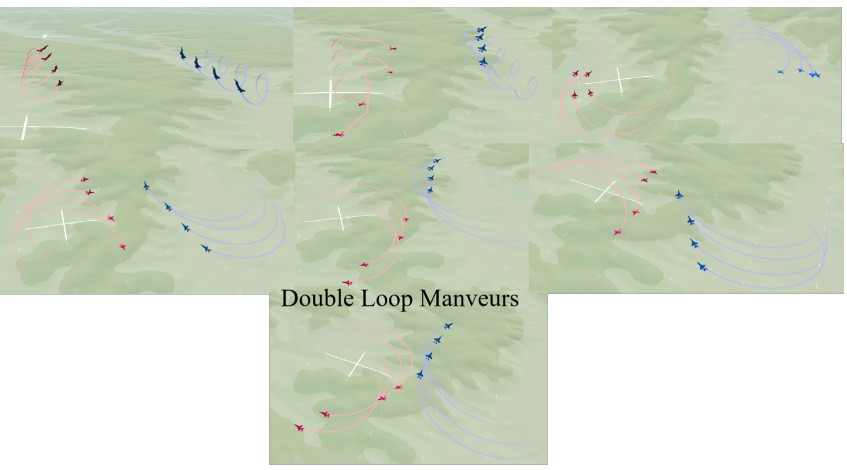

Figure 10: Snapshot of ours vs. ours.

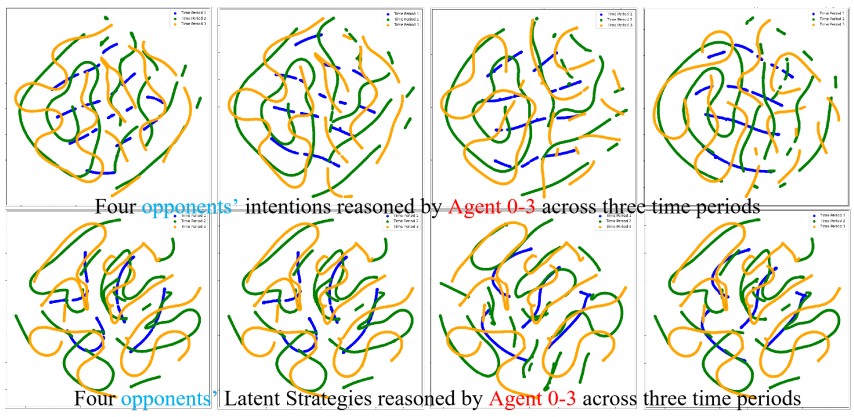

Figure 11: Mental states of opponents reasoned by red team agents.

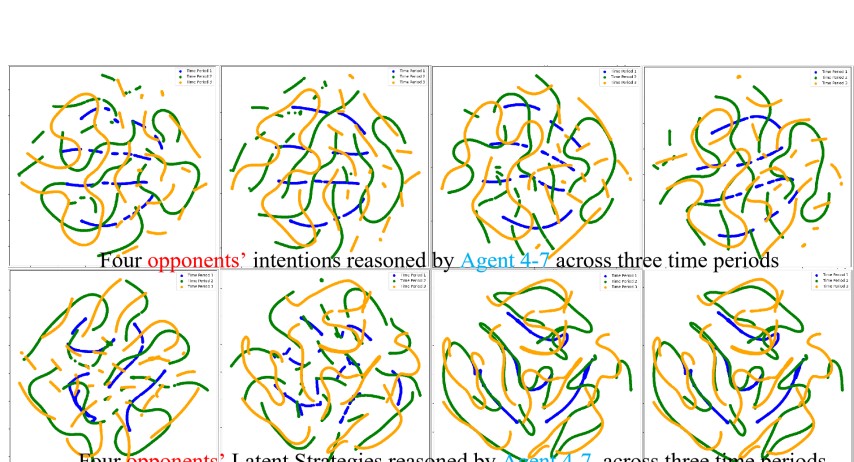

Figure 12: Mental states of opponents reasoned by blue team agents.

