# OpenReview forum: "H2IL-MBOM: A Hierarchical World Model Integrating Intent and Latent Strategy as Opponent Modeling in Multi-UAV Game"
_ICLR.cc/2025/Conference — ICLR 2025 Conference Withdrawn Submission_

### Official Review · Reviewer_vYGG · 2024-11-01

**Soundness:** 3
**Presentation:** 1
**Contribution:** 2
**Rating:** 5
**Confidence:** 4

**Summary:**

This paper presents H2IL-MBOM, a hierarchical model for opponent modeling in multi-agent reinforcement learning, particularly in air combat scenarios. H2IL-MBOM combines high-level intention inference with low-level strategy prediction to address non-stationary dynamics in mixed cooperative-competitive settings. Integrated into the PPO framework, this model achieves enhanced accuracy and interpretability, showing improved performance over baseline methods in simulations.

**Strengths:**

1.	The approach of modeling opponents through world models in air combat scenarios is innovative.

2.	H2IL-MBOM models opponents based on observational data, providing a useful approach for scenarios such as air combat, where direct access to the opponent's precise actions and states is unavailable.

3.	Comprehensive experiments in Gym-Jsbsim demonstrate the method’s significant performance advantages over model-free MARL and other opponent modeling methods, including ablation studies that confirm module effectiveness. Sufficient details of the experimental implementation are also provided.

4.	In the experimental section, Figure 3 and Appendix A.13 effectively demonstrate and validate that the proposed method can capture changes in opponent intentions in the air combat environment.

**Weaknesses:**

1.	The paper’s expression and presentation lack clarity; the authors provide numerous equations for various modules, making it difficult to smoothly understand the intent and overall functionality of the H2IL-MBOM framework. The extensive use of abbreviations also confuses readers. Figure 1, intended as an overview of H2IL-MBOM, includes excessive module details, which makes it challenging for readers to grasp the authors' main ideas. Consider breaking it down or omitting unnecessary details.

2.	Although applying the method of modeling opponents through world models in the air combat environment is innovative, I still hope the authors can conduct comparative experiments in more multi-agent adversarial environments, such as Google Football, and compare against additional baselines to demonstrate the advantages of the proposed method, especially since there is relatively less existing work in MARL for air combat environments.

3.	There are still some type errors, such as in the first paragraph of Section 3, where it says, "and using these these predictions along with observations to inform decision-makings."

4.	The authors do not provide a detailed analysis or validation of the effectiveness of the opponent model in the methods and experimental sections.

**Questions:**

1.	Is H2IL-MBOM equally effective in other multi-agent tasks and environments?

2.	Is there a rationale for the design of the action space, state space, and reward function used in the reinforcement learning (RL) framework?

---

### Official Review · Reviewer_QuNh · 2024-11-03

**Soundness:** 3
**Presentation:** 2
**Contribution:** 2
**Rating:** 3
**Confidence:** 4

**Summary:**

The paper introduces H2IL-MBOM, a hierarchical world model that integrates intent and latent strategy for opponent modeling in multi-UAV games. It addresses challenges in mixed cooperative-competitive scenarios by enabling dynamic prediction of opponents' intentions and strategies. The proposed MSOAR-PPO algorithm allows for real-time inference of adversaries' strategies and intentions, facilitating rapid adaptation to changes in opponents' behaviors. The method's effectiveness is demonstrated through comparisons with state-of-the-art methods in multi-agent air-combat simulations, showing superior performance and generalization ability. The paper concludes that H2IL-MBOM enhances decision-making in complex, dynamic environments by accurately capturing opponents' mental states and their evolving strategies.

**Strengths:**

- The proposed method demonstrates superior performance when compared to state-of-the-art model-free reinforcement learning and opponent modeling methods. It effectively captures the changing behavior patterns of opponents and exhibits strong generalization capabilities in multi-agent close-range air-combat environments with missiles.
- The H2IL-MBOM, coupled with the MSOAR-PPO algorithm, enables dynamic and interactive prediction of multiple intentions and latent strategies. This allows for real-time adaptation to changes in opponents' intentions and strategies, addressing the non-stationarity issue in multi-agent interactions and enhancing decision-making processes.

**Weaknesses:**

- Lack of novelty. Modeling others and world dynamics for multi-agent reinforcement learning has been widely explored in previous works[1,2,3]. It is necessary to justify the novelty of the proposed hierarchical framework. Although previous works do not apply to the multi-UAV game, I can not see any additional challenge introduced in the game.
- Most of the baselines are out-of-date, e.g., MADDPG, MAPPO. It is necessary to compare stronger baselines with the SOTA opponent modeling methods that were introduced for general multi-agent games.
- Generalization. The generalization of the learned model to different numbers of agents/opponents and unseen behaviors during test time is not evaluated. The paper primarily focuses on air-combat scenarios. It is not clear how well the proposed methods would generalize to other types of multi-agent environments with different dynamics and objectives.
- The hierarchical model proposed is complex, which could limit its scalability and applicability.


References:
[1] Proactive Multi-Camera Collaboration for 3D Human Pose Estimation, ICLR 2023

[2] Fast Peer Adaptation with Context-aware Exploration, ICML 2024

[3] Greedy when sure and conservative when uncertain about the opponents, ICML 2022

**Questions:**

- How to extend the framework to handle the visual observation for real-world applications?
- Can you show some videos about the simulation and the learned policy?
- Is the model robust to the different scales of the population of agents, e.g. 10 vs. 10?

**Details Of Ethics Concerns:**

Military Applications and Escalation: The model could be used to enhance military drone technologies, potentially leading to more efficient and lethal autonomous weapon systems. This raises concerns about the escalation of armed conflict and the dehumanization of warfare.

---

### Official Review · Reviewer_P9KM · 2024-11-09

**Soundness:** 2
**Presentation:** 1
**Contribution:** 2
**Rating:** 3
**Confidence:** 3

**Summary:**

This paper presents a multi-agent model-based reinforcement learning framework for close-range air combat. Compared to prior work, this paper employs a more realistic observation model in which agents cannot observe private state information from other agents. The main contribution of the paper is a data-driven two-level latent variable model. The high-level model learns a latent space for "intentions," and the low-level model learns another for "strategies." The forward world model consists of models for intentions and strategies and how they affect future states/observations. These models are parameterized by Transformers, similar to prior work on TSSM.

The authors employ a self-play setting to evaluate the RL agent performance in a simulated air combat environment. The main results demonstrate that the proposed method achieves higher rewards than relevant model-free and model-based RL baselines. The results suggest that the novel hierarchical modeling approach helps more accurately predict the interleaving dynamics in a multi-agent environment.

**Strengths:**

- The main algorithm is a sophisticated approach to solving a challenging, practical multi-agent control problem.
- The authors include in-depth derivation and detailed algorithms in the appendix.
- The algorithm outperforms various model-free and model-based RL baselines.

**Weaknesses:**

The presentation of the paper needs improvement:
- The paper's key contribution is modeling the dynamics of "intentions" and "strategies," but I don't see a clear definition. Are they just two generic latent spaces the authors have assigned names to? What is a mental state?
- The paper is swamped with rather random-looking abbreviations. These don't flow well in sentences, making the method section difficult to follow.
- The experiment settings are not communicated clearly (see questions).
- The result figures are noisy. The authors should consider running multiple seeds to visualize the average trend. Also, the text in the figures is too small, and the captions are not informative at times.
- Equation 1 is outside the paper margin.

Overall, I often find it hard to distinguish if a statement is a motivation, a hypothesis, or some standard definition from prior work (e.g., section 3.1).

**Questions:**

- How are the plotted rewards computed? My understanding is that the authors use self-play during training. Then, how is the policy performance evaluated? Are the baseline methods compared on the same opponent team?
- Does each agent make decisions independently? I don't think the MDP formulation is appropriate here because not everything is observable. Also, the MDP seems to describe the entire simulation state but not from individual agents' point of view.

---

### Note · Authors · 2024-11-14

**Comment:**

Respond to reviewer 1
1. The definition of intentions and latest strategies are presented in lines 207-209, "the opponent's evolving intention directly reflects the changes in the opponents' trajectories, while their evolving strategy influcences the trajectories of the alliance agents". The definition of mental state is detailed in lines43, and we suggest that the reviewer carefully read the paragraphs of lines 42-63, 98-116 and 191-215. The reviewer seems to be unfamiliar with the relevant concepts and this field.
2.  As sentence of lines 263 and 356-357 say, alliances and opponents are equipped with the same H2IL MBOM, and they learn independently **without adopting our old strategy through self play**. The pseudocode in lines 707 can also verify this.  And in the experiment, we also carefully described our experimental setup, as lines 407-411 states, "It is notifying that we use two,,,,,,, one is where, ,,,,,,, and the other is,,,,,,".  Moreover, the self play method was compared in Figure 2 (d).
3. All methods are against the same group of opponents, and the reward is the average episode return of all agents on the same team.
4. Our MSOAR-PPO are based on th ippo method，which is different from Mappo and IPPO . Because the observations of opponents that can be not accessed in reality，we neither use global observations nor own respective local observations in the critic network, but instead use observable teammates' observations. Additionally, in the actor network, we combine local observations with reasoning intentions and strategies to make decisions. **And our input is not the state, but rather local observations relative to the state of our teammates and opponents within a limited observation range.** In other words, the assumptions about global observation for critic network of mappo do not match the actual situation. It seems that the reviewers are not very familiar with IPPO, MDP, and CTDE, especially the relationship between IPPO and MDP.

Respond to reviewer 2
1. The references 1-3 cited first are not quite the same as what we are concerned about, especially the mutual inference of multiple agents between two teams.
2. The challenge of this article has been elaborated in detail in lines 42-63 of the introduction.
3. We valide the effectiveness of opponent model method  such as Rommeo， PR2， TDOM-AC， and AORPO in Figure2 , and there are few opponent modeling methods for multi-agent intent inference. In some articles, although it is a multi-agent setting, other agents are mainly treated as opponents, and the interested agent is used to reason about these opponents, rather than interactive reasoning between teammates for all agents. And our method takes into account that the opponent has the same reasoning ability, but it is not simply a self play approach. In addition, as lines 46-58 say, other methods either use the opponent's private information as label data or do not consider the continuous mutual influence of the intentions, strategies, and trajectories of both parties.
4. The real-world applications can also be discussed in the section limits of appendices, and the visualized trajectories can be found in section A13 of appendices.
5. The generalization experiments of 10vs10 are presented in the section "testing results" of appendices.

Respond to reviewer 3

1.  As the title suggests, we are concerned with multi drone game tasks, and articles on intent inference in the field of autonomous driving will not be validated in football experiments. This additional requirement for irrelevant experiments seems unfair to us. Moreover, our experiment was conducted in the mixed cooperative-competitive scenario, and both parties had the ability to reason about their opponents. The biggest difference between this and SMAC and football environments is that their opponents use built-in AI, i.e. pure collaborative scenarios, while our method, as lines408 stated, does not use built-in AI. These environments are not aligned with our mission objectives.

Nevertheless, we believe our H2IL-MBOM is equally effective in other multi-agent tasks and environments.

2.  The meaning of “using these these predictions along with observations to inform decision-makings.”is to make decisions based on the observation and speculated intentions and strategies.


3.  We valide the effectiveness of opponent model method  such as Rommeo， PR2， TDOM-AC， and AORPO in Figure2. And the effectiveness of our method for opponent intention and strategy inference was also verified and analyzed through tsne distribution in Figure 3 (c) - (d).

4. This environment[1] is open source and has become a widely used benchmark for drone gaming, with its observation space, action space, and rewards widely recognized.

[1] Qihan Liu, Xiaoteng Ma, et. al, “Light Aircraft Game: A lightweight, scalable, gym-wrapped aircraft competitive
environment with baseline reinforcement learning algotithms.” https: //github.com/liuqh16/CloseAirCombat, 2022.

**Withdrawal Confirmation:**

I have read and agree with the venue's withdrawal policy on behalf of myself and my co-authors.